# Treacle controls the nucleolar response to rDNA breaks via TOPBP1 recruitment and ATR activation

Clémence Mooser[1,5], Ioanna-Eleni Symeonidou[1,5], Pia-Amata Leimbacher[1], Alison Ribeiro[1], Ann-Marie K. Shorrocks[2,3], Stephanie Jungmichel[4], Sara C. Larsen [4], Katja Knechtle[1], Arti Jasrotia[1], Diana Zurbriggen[1], Alain Jeanrenaud[1], Colin Leikauf [1], Daniel Fink[1], Michael L. Nielsen [4], Andrew N. Blackford [2,3] & Manuel Stucki [1]*

Induction of DNA double-strand breaks (DSBs) in ribosomal DNA (rDNA) repeats is associated with ATM-dependent repression of ribosomal RNA synthesis and large-scale reorganization of nucleolar architecture, but the signaling events that regulate these responses are largely elusive. Here we show that the nucleolar response to rDNA breaks is dependent on both ATM and ATR activity. We further demonstrate that ATM- and NBS1-dependent recruitment of TOPBP1 in the nucleoli is required for inhibition of ribosomal RNA synthesis and nucleolar segregation in response to rDNA breaks. Mechanistically, TOPBP1 recruitment is mediated by phosphorylation-dependent interactions between three of its BRCT domains and conserved phosphorylated Ser/Thr residues at the C-terminus of the nucleolar phosphoprotein Treacle. Our data thus reveal an important cooperation between TOPBP1 and Treacle in the signaling cascade that triggers transcriptional inhibition and nucleolar segregation in response to rDNA breaks.

[1] Department of Gynecology, University of Zurich, Wagistrasse 14, CH-8952 Schlieren, Switzerland. [2] Department of Oncology, Medical Research Council Weatherall Institute of Molecular Medicine, University of Oxford, John Radcliffe Hospital, Oxford OX3 9DS, UK. [3] Cancer Research UK/Medical Research Council Oxford Institute for Radiation Oncology, University of Oxford, Oxford OX3 7DQ, UK. [4] The Novo Nordisk Foundation Center for Protein Research, University of Copenhagen, Faculty of Health and Medical Sciences, Bledgamsvej 3B DK-2200, Copenhagen, Denmark. [5] These authors contributed equally: Clémence Mooser, Ioanna-Eleni Symeonidou. *email: manuel.stucki@uzh.ch

D ouble-strand breaks (DSBs) are highly toxic lesions that need to be repaired accurately in order to avoid chromosomal instability, a hallmark of various human disorders, including cancer. Mammalian cells respond to DSBs by activating a complex network of signaling pathways generally referred to as the DNA damage response (DDR)[1], which regulates various cellular processes such as the induction of cell-cycle checkpoints, apoptosis or senescence and the coordination of DNA repair pathway choice.

Central players in the cellular response to DSBs are the phosphoinositide 3-kinase (PI3K)-related protein kinases (PIKKs), including ataxia-telangiectasia mutated (ATM) and ATM and RAD3-related (ATR). ATM functions mainly in response to DSBs while ATR is activated by a much wider range of genotoxic insults. Both ATM and ATR require specific cofactors for their recruitment and optimal activation. The MRE11/RAD50/NBS1 (MRN) complex is required for ATM and ATR activation in response to DSBs[2–5]. ATR is recruited to sites of DNA damage via its interacting protein ATRIP that binds to ssDNA coated with the ssDNA-binding protein RPA (RPA-ssDNA)[6]. Such RPA-ssDNA is generated by nucleolytic processing of DNA lesions or by the uncoupling of DNA polymerases and helicases during DNA replication[7]. Several additional cofactors are required for the optimal activation of ATR at sites of DNA damage. Among these, the best studied is TOPBP1, which contains an ATR activation domain (AAD) that stimulates ATR kinase activity[8].

DSBs also induce transcriptional repression[9–14]. This effect has first been described in the nucleoli where the ribosomal gene (rDNA) arrays that make up for the highest transcribed sites in the human genome are located[9]. In human cells, ~300 rDNA repeats are distributed among nucleolar organizer regions (NORs). The NORs are located on the short arms of the five acrocentric chromosomes and encode the precursor ribosomal RNA (pre-rRNA) that is synthesized by the RNA polymerase I (Pol I) transcription machinery. Nucleoli are composed of three distinct subdomains that reflect the stages of ribosome biogenesis. Fibrillar centers (FCs) contain pools of transcription factors and non-transcribed rDNA sequences. Pol I transcription takes place at the interface between FCs and the surrounding dense fibrillar component (DFC) where early processing of the synthesized pre-rRNA occurs. FCs and DFC are embedded in the granular component (GC) where late processing of pre-rRNAs occurs, yielding mature rRNAs that together with the ribosomal proteins assemble into ribosome subunits[15].

The induction of breaks in the rDNA repeats triggers rapid ATM-dependent shutdown of Pol I transcription in the nucleoli[9]. Transcriptional repression is accompanied by a large-scale structural alteration of the nucleoli whereby the rDNA repeats along with associated proteins translocate from inside the nucleoli into the nucleolar periphery, where they accumulate in focal structures that are called nucleolar caps[16–19]. These structural changes—often referred to as nucleolar segregation—are also induced by inhibition of Pol I transcription by small-molecule inhibitors, thus suggesting that transcriptional repression is the underlying cause of nucleolar segregation[20]. While nucleolar segregation has been proposed to facilitate the repair of rDNA breaks[16,17,19], the mechanism by which ATM signaling suppresses rRNA transcription and the functional implication of transcriptional repression and nucleolar segregation have not yet been studied in detail.

We and others recently discovered that NBS1, a subunit of the MRN complex, is rapidly and transiently recruited in the nucleoli in response DSBs[21,22]. This recruitment event is dependent on a direct phosphorylation-dependent interaction between the N-terminal FHA/BRCT region of NBS1 and Treacle (also

called TCOF1), a low complexity nucleolar phosphoprotein whose gene is mutated in Treacher-Collins syndrome, a congenital disorder of craniofacial development[23]. Even though nucleolar localization of NBS1 has been associated with transcriptional repression of the rDNA repeats, the exact mechanism by which NBS1 contributes to the shutdown of Pol I transcription remains elusive.

Here we show that transcriptional repression and nucleolar segregation in response to targeted induction of DSBs in the rDNA repeats depend on both ATM and ATR activity. We furthermore show that Treacle recruits the ATR activator TOPBP1 in the nucleoli in an ATM- and NBS1-dependent manner. Mechanistically, TOPBP1 is recruited through direct phosphorylation-dependent interactions with the C-terminal region of Treacle. We propose that Treacle orchestrates the nucleolar response to rDNA breaks primarily through recruiting the two key adaptors NBS1 and TOPBP1 in the nucleoli, both of which are implicated in ATR activation.

## Results

**ATR-dependent Pol I inhibition and nucleolar segregation.** In order to follow the dynamics of the cellular response to rDNA breaks in real time, we generated U2OS cell lines stably expressing low levels of green fluorescence protein (GFP)-tagged Treacle (Supplementary Fig. 1a). Characterization of these cell lines showed that GFP-Treacle exhibited the same subcellular localization as the endogenous Treacle, concentrating at the FC and DFC within the nucleoli (Fig. 1a). To induce targeted breaks in the rDNA repeats, we adopted the I-Ppo1 mRNA transfection approach previously described[16]. I-Ppo1 recognizes a sequence within the 28S portion of the rDNA repeat as well as a few sites elsewhere in the genome[24]. Live-cell microscopy was used to monitor changes in GFP-Treacle localization and changes in nucleolar morphology over time. Upon induction of breaks in the rDNA repeats, GFP-Treacle foci rapidly fused to form a few bright centers within the nucleoli before they migrated in the nucleolar periphery. At the same time, the nucleoli changed from irregularly shaped to round. By 120 min post I-Ppo1 mRNA transfection, the majority of nucleoli in the transfected cell population had completely segregated (Fig. 1a and Supplementary Movie 1).

We next asked if the DDR is implicated in the translocation of GFP-Treacle from inside the nucleoli to the nucleolar periphery in response to rDNA breakage. Surprisingly, we found that nucleolar segregation is not only blocked by ATM inhibitors, but also by ATR inhibitors (Supplementary Fig. 1b). We next assessed the localization of endogenous Treacle in response to I-Ppo1 mRNA transfection by quantitative 3D confocal microscopy (Supplementary Fig. 2a). These studies confirmed that migration of endogenous Treacle to the nucleolar periphery as well as changes in the nucleolar volume and sphericity in response to rDNA breaks are dependent on both ATM and ATR activity (Fig. 1c, d, Supplementary Fig. 2c). We observed that nucleolar segregation is blocked by ATR inhibitors in the majority of cells, irrespective of the cell-cycle stage (Supplementary Fig. 2d). The differences between control cells and inhibitor treated cells were not caused by variability in the I-Ppo1 expression levels nor did inhibitor treatment affect I-Ppo1 translation (Supplementary Fig. 3a–c). We next investigated ATM- and ATR recruitment to sites of I-Ppo1-induced DSBs in the nucleoli. Both ATM and ATR first accumulated inside of the nucleoli and at later timepoints, localized to nucleolar caps (Fig. 1e, Supplementary Fig. 2b). We also assessed ATM and ATR downstream signaling in response to I-Ppo1 transfection. CHK2 was efficiently phosphorylated on

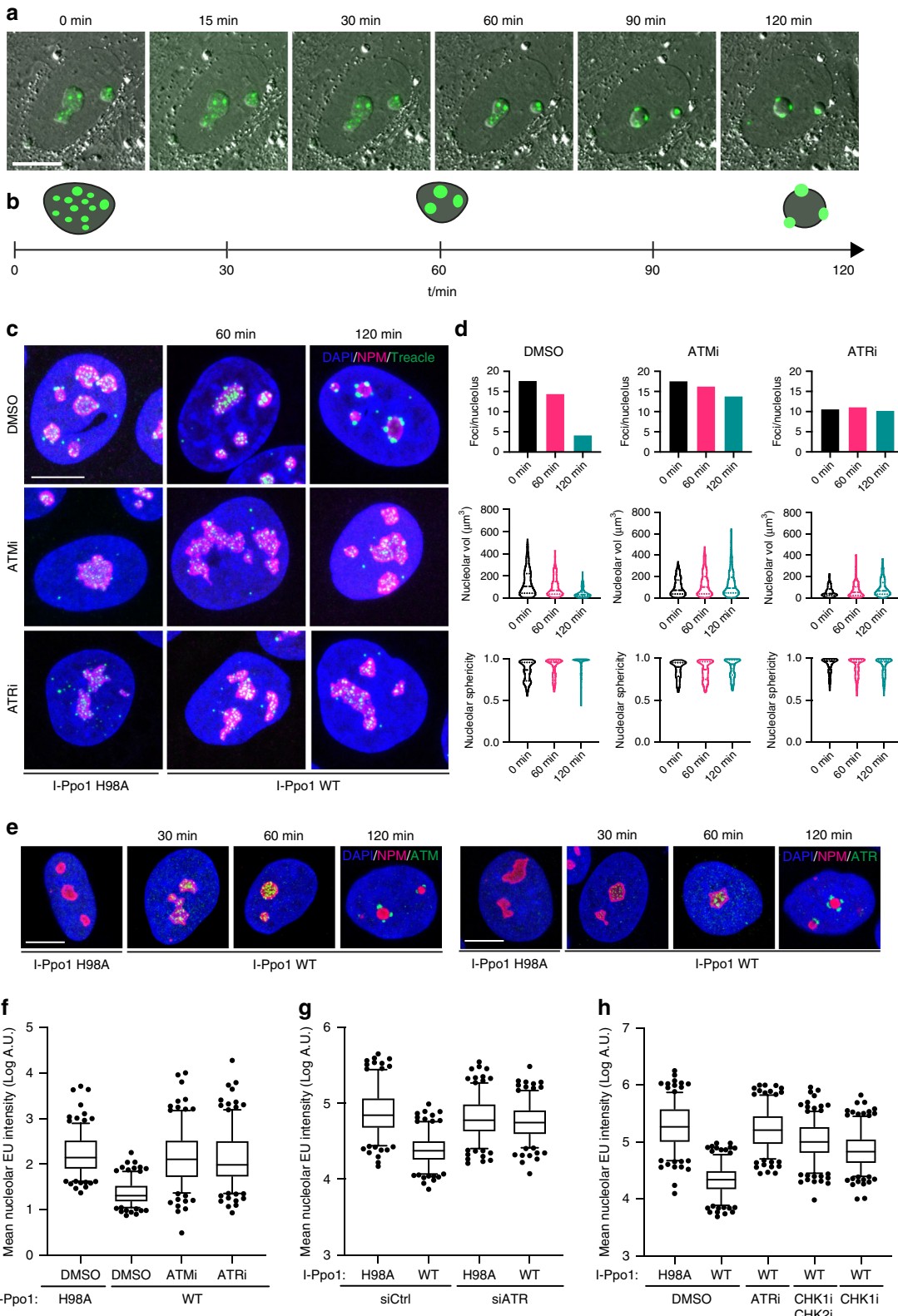

Thr68 in an ATM-dependent manner, and CHK1 was phosphorylated on both Ser317 and Ser345 in an ATR-dependent manner (Supplementary Fig. 1c).

Next, we measured transcriptional activity within the nucleoli in response to rDNA breaks by 5-ethynyl uridine (EU) RNA labeling followed by Click-iT chemistry and by quantitative real-time PCR (qRT-PCR). Both assays showed strong ATM and ATR

dependency of Pol I transcriptional inhibition in response to targeted induction of rDNA breaks by I-Ppo1 and in response to ionizing radiation (Fig. 1f, g and Supplementary Fig. 1d, e). Notably, CHK1 and CHK2 inhibition by two small-molecule inhibitors also significantly compromised transcriptional inhibition, albeit somewhat less efficiently than ATR inhibition (Fig. 1h).

**Fig. 1 ATR-dependent Pol I inhibition and nucleolar segregation. a** Time-lapse microscopy of U2OS cells expressing low levels of GFP-tagged Treacle, after transfection with I-Ppo1. **b** Timeline of the morphological changes of the nucleoli after I-Ppo1 transfection. **c** Morphological changes of the nucleoli after I-Ppo1 transfection in ATMi (KU-55933) and ATRi (VE-821) treated cells. Displayed are maximum intensity projections of confocal z-stacks. NPM stands for nucleophosmin **d** Quantification of Treacle foci number per nuceolus, nucleolar volume and nucleolar sphericity by segmentation of 3D reconstituted z-stacks. DMSO/0 min $n = 262$, DMSO/60 min $n = 239$, DMSO/120 min $n = 264$, ATMi/0 min $n = 174$, ATMi/60 min $n = 166$, ATMi/120 min $n = 179$, ATRi/0 min $n = 208$, ATRi/60 min $n = 218$, ATRi/120 min $n = 201$ independent cells. Bars represent means, dotted lines within the violin plots represent median and quartiles. **e** ATM and ATR recruitment to sites of nucleolar DSBs in response to I-Ppo1 transfection. Displayed are maximum intensity projections of confocal z-stacks. NPM stands for nucleophosmin. **f** Quantification of nucleolar EU incorporation after I-Ppo1 transfection in I-Ppo1 H98A/DMSO ($n = 215$), DMSO ($n = 205$), ATMi (KU-55933, $n = 206$) and ATRi (VE-821, $n = 219$) treated cells. **g** Quantification of nucleolar EU incorporation in siCtrl I-Ppo1 H98A and WT ($n = 203$) and siATR I-Ppo1 H98A and WT ($n = 210$) treated cells. **h** Quantification of nucleolar EU incorporation after I-Ppo1 H98A ($n = 229$) and I-Ppo1 WT transfection in control (DMSO, $n = 231$), ATRi (VE-821, $n = 237$), CHK1i/CHK2i (AZD7762, $n = 246$) and CHK1i (GDC-0575, $n = 255$) treated cells. **f–h** Boxes represent the 25–75 percentile range with median and whiskers represent the 5–95 percentile range. Data points outside of this range are shown individually. All scalebars = 10 μm. Source data are provided as a Source Data file.

In summary, these data reveal that transcriptional inhibition and nucleolar segregation in response to rDNA breakage require both ATM and ATR activity throughout the cell cycle.

**NBS1 regulates the nucleolar response to rDNA breaks**. As we and others previously showed that NBS1 interacts with Treacle and is rapidly recruited in the nucleoli in response to DSBs in an ATM- and Treacle-dependent manner[21,22], we next wished to determine if NBS1 is also implicated in the nucleolar response to I-Ppo1 expression. To this end, we targeted exon 2 of NBS1 in U2OS cells using CRISPR-Cas9 and isolated several clones with no detectable expression of full-length NBS1 (Supplementary Fig. 4a–c). One clone was subsequently analyzed in detail. Similar to cell lines derived from Nijmegen breakage syndrome patients, NBS1 knock-out U2OS cells showed increased radio-sensitivity, displayed defects in G2/M checkpoint activation, DSB resection and ATR activation in response to ionizing radiation (IR), and had MRE11 predominantly localized in the cytoplasm (Supplementary Fig. 5a–f). As it was recently shown that the C-terminal MRE11-interacting region in NBS1 is sufficient for viability and ATM activation[25], we suspected that the NBS1 knock-out U2OS cells may express a hypomorphic C-terminal fragment of NBS1. Indeed, the NBS1 mRNA is overexpressed in these cells and they express low levels of an ~40 kDa protein that is responsive to NBS1 mRNA knockdown by siRNA (Supplementary Fig. 5g, h). Based on these findings, we concluded that these cells most likely express low levels of a truncated NBS1 protein lacking the N-terminus. Therefore, we henceforth refer to these cells as NBS1ΔN.

We next addressed the role of NBS1 in transcriptional repression and nucleolar segregation in response to I-Ppo1 transfection. No nucleolar segregation was evident in NBS1ΔN cells within the first 2 h after transfection with I-Ppo1. (Fig. 2a). In addition, Pol I inhibition upon I-Ppo1 expression was compromised in NBS1ΔN cells (Fig. 2b). We next constructed NBS1ΔN cell lines stably expressing full-length NBS1, tagged at the C-terminus with the bright monomeric GFP variant mNeon-Green (mNG). Time-lapse microscopy of these cells revealed normal nucleolar segregation in response to I-Ppo1 expression. In addition, we consistently observed that NBS1-mNG is recruited into the nucleoli prior to nucleolar segregation (Fig. 2c, d, Supplementary Movie 2). NBS1-mNG co-localized with Treacle in the nucleoli and the introduction of point mutations in the N-terminal FHA and BRCT domains of NBS1 that have previously been shown to abrogate their ability to interact with phosphorylated proteins (R28A, K160M)[26], compromised nucleolar recruitment of NBS1 (Fig. 2e). Since NBS1 contacts Treacle via its FHA/BRCT domains[21], these data suggest that direct interaction with Treacle may also mediate its recruitment in the nucleoli. Indeed, depletion of endogenous Treacle by siRNA strongly reduced

NBS1 nucleolar recruitment in response to I-Ppo1 expression (Fig. 2f). Treacle-mediated recruitment of NBS1 was prevented by ATM and ATR inhibition (Supplementary Fig. 6a). Since we also observed MRE11 recruitment in the nucleoli after I-Ppo1 transfection (Supplementary Fig. 6b), we propose that NBS1 acts in the nucleolar response to rDNA breaks in the context of the entire MRN complex.

In summary, these results confirm that the MRN complex is an essential upstream component of the signaling pathways that regulate transcriptional repression and nucleolar segregation in response to rDNA breaks[9,21].

**rDNA resection occurs downstream of ATR activation**. The MRN complex has been proposed to act upstream of ATR activation in response to DSBs through its essential role in DNA end resection, which produces ssDNA, the main signal recognized by ATR[4,5,27]. We therefore considered the possibility that ATM- and MRN-dependent resection of rDNA breaks may be required for ATR activation in the nucleoli. I-Ppo1 expression triggered the accumulation of the ssDNA binding protein RPA within nucleolar caps (Fig. 3a). In addition, RPA2, phosphorylated on Ser4 and Ser8 (RPA2 pS4/8), an established marker for resection, was also enriched within nucleolar caps (Supplementary Fig. 7a). At early time points after I-Ppo1 transfection, we observed RPA foci also inside the nucleoli, suggesting that DNA end resection occurs already before the broken rDNA repeats have moved out of the nucleoli (Fig. 3a). It was proposed that breaks in the rDNA repeats are resected throughout the cell cycle[16]. However, we found strong cell cycle dependency of RPA foci formation within the first 2 h after I-Ppo1 transfection, with only Cyclin A-positive S/G2-phase cells showing significant accumulation of RPA in the nucleolar periphery (Fig. 3b, c and Supplementary Fig. 7b, c). In NBS1ΔN cells, no RPA foci formed within the first 2 h after I-Ppo1 transfection (Fig. 3d, e). RPA pS4/8 foci were also strongly reduced in NBS1ΔN cells and by the MRE11 inhibitor Mirin, indicating that resection of broken rDNA is dependent on the nuclease activity of MRN complex (Supplementary Fig. 7a).

Surprisingly, we observed that treatment of cells with ATM and ATR inhibitors prior to I-Ppo1 transfection strongly abrogated RPA foci formation, suggesting that DNA end resection is not required for ATR activation in the nucleoli (Fig. 3f, g). In support of this, depletion of the DNA end resection factor CtIP failed to compromise nucleolar segregation, even though it strongly inhibited RPA foci formation in response to I-Ppo1 expression (Supplementary Fig. 7d).

These findings thus reveal a difference in the regulation of DNA end resection between DSBs induced in the rDNA repeats as compared to DSBs induced elsewhere in the genome that do not require ATR activity for resection.

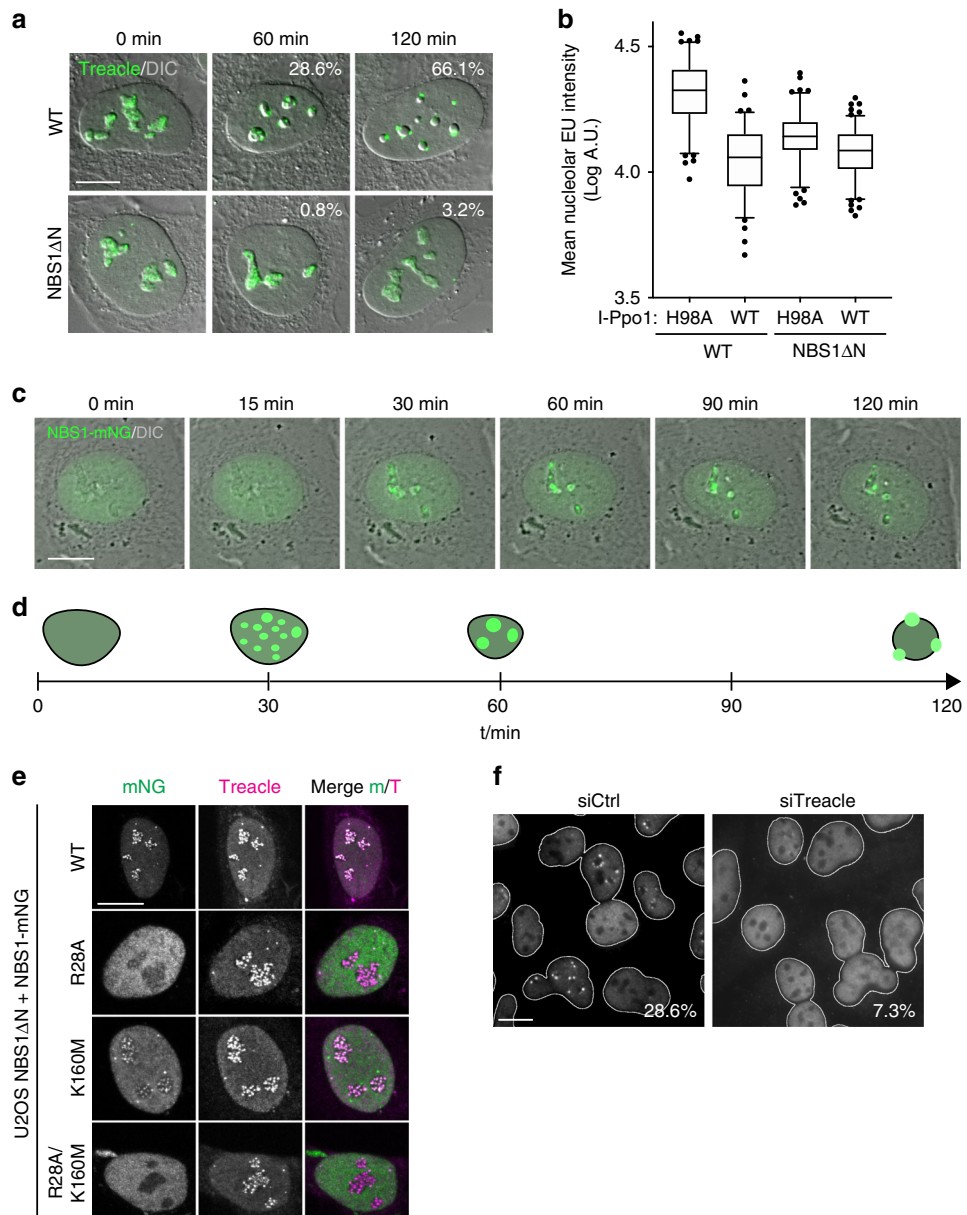

**Fig. 2 NBS1 regulates the nucleolar response to rDNA breaks. a** Timecourse of endogenous Treacle localization after I-Ppo1 transfection in normal U2OS cells and NBS1ΔN cells. **b** Quantification of nucleolar EU incorporation after I-Ppo1 H98A (*n* = 111) and I-Ppo1 WT (*n* = 87) transfection in control U2OS and after I-Ppo1 H98A (*n* = 116) and I-Ppo1 WT (*n* = 124) transfection in NBS1ΔN cells (boxes represent the 25–75 percentile range with median and whiskers represent the 5–95 percentile range. Data points outside of this range are shown individually). **c** Time-lapse microscopy of U2OS cells expressing mNG-tagged full-length NBS1 (wild type), after transfection with I-Ppo1. **d** Timeline of NBS1-mNG recruitment in relation to morphological changes in the nucleoli after I-Ppo1 transfection. **e** Confocal microscopy of NBS1-mNG wild type and point mutants in the NBS1ΔN background. **f** Localization of endogenous NBS1 2 h after I-Ppo1 transfection in siCtrl and siTreacle-treated cells (percentages of cells with >2 NBS1 caps are indicated; one of two experiments is shown). All scalebars = 10 μm. Source data are provided as a Source Data file.

**TOPBP1 recruitment in response to rDNA breaks**. Having established that DNA end resection is not required for nucleolar segregation in response to I-Ppo1 expression and occurs downstream of ATR activation, we next wished to investigate the mechanism of ATR activation in the nucleoli. The ATR activator TOPBP1 was previously implicated in nucleolar stress responses and its overexpression was shown to trigger ATR-dependent Pol I inhibition and nucleolar segregation even in the absence of rDNA damage[28]. Upon induction of rDNA breaks by I-Ppo1, TOPBP1 first accumulated inside of the nucleoli where it co-localized with Treacle. At later timepoints, after nucleolar segregation had taken place, TOPBP1 accumulated in nucleolar caps (Fig. 4a).

We next asked if TOPBP1 recruitment in the nucleoli is dependent on NBS1, ATM and ATR activity. Only few NBS1ΔN cells contained nucleolar TOPBP1 foci, and expression of wild-type NBS1 but not FHA/BRCT mutated NBS1 rescued this defect (Fig. 4b). In addition, TOPBP1 accumulation in nucleolar caps was also blocked by pre-treatment of cells with either ATM or ATR inhibitors (Fig. 4c, d). Interestingly though, TOPBP1 foci did form within the nucleoli of a substantial percentage of ATR-inhibited U2OS cells, suggesting that not the initial recruitment of TOPBP1 but rather its subsequent accumulation in the nucleolar periphery is dependent on ATR activity (Fig. 4c).

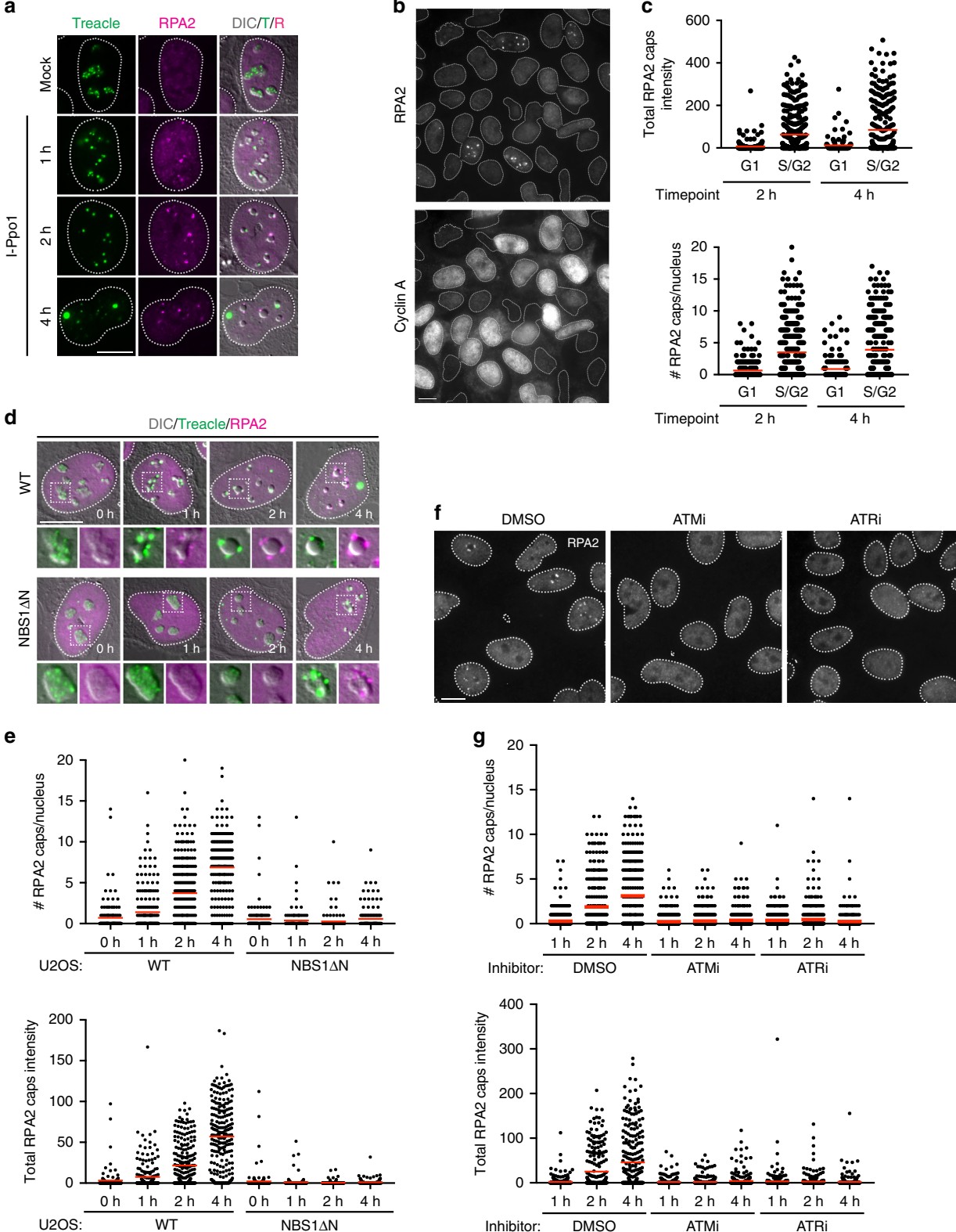

**Fig. 3 rDNA resection occurs downstream of ATR activation. a** Timecourse of RPA2 foci formation in U2OS cells after I-Ppo1 transfection. **b** Timecourse of RPA2 foci formation in cells counterstained for Cyclin A. **c** Quantification of the experiment in **b** (all data points shown, red bars represent mean). **d** Timecourse of RPA2 foci formation in U2OS WT and NBS1ΔN cells after I-Ppo1 transfection. **e** Quantification of the experiment in **d** (all data points shown, red bars represent mean). **f** Timecourse of RPA2 foci formation in U2OS cells treated with ATMi (KU-55933) and ATRi (VE-821) after I-Ppo1 transfection. **g** Quantification of the experiment in **f** (all data points shown, red bars represent mean). All scalebars = 10 μm. Source data are provided as a Source Data file.

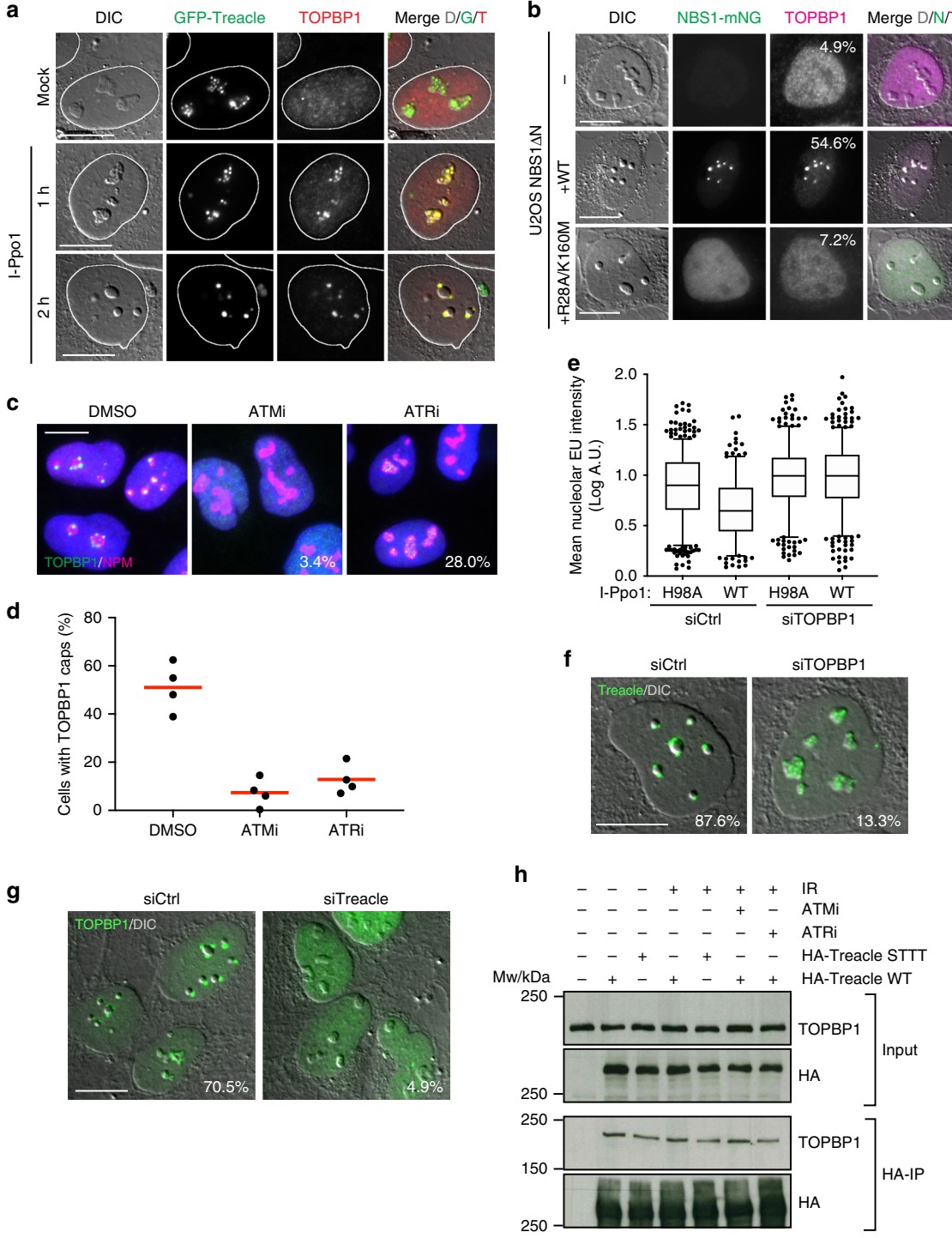

**Fig. 4 TOPBP1 recruitment in response to rDNA breaks. a** Timecourse of TOPBP1 localization in GFP-Treacle expressing U2OS cells after I-Ppo1 transfection. **b** TOPBP1 localization 2 h after I-Ppo1 transfection in NBS1ΔN cells and NBS1ΔN cells complemented with wild-type and mutant NBS1-mNG (percentage of cells with > 2 TOPBP1 caps are indicated; one of two experiments is shown). **c** TOPBP1 localization 2 h after I-Ppo1 transfection in ATMi- and ATRi-treated cells versus vehicle (DMSO)-treated cells (percentage of cells with TOPBP1 foci inside of the nucleoli are indicated; one of two experiments is shown). **d** Quantification of cells with TOPBP1 caps after ATMi and ATRi treatment versus vehicle (DMSO) treatment (red bars represent mean). **e** Quantification of nucleolar EU incorporation after I-Ppo1 H98A (n = 692) and I-Ppo1 WT (n = 649) transfection in control U2OS and after I-Ppo1 H98A (n = 450) and I-Ppo1 WT (n = 521) transfection in siTOPBP1-treated cells (boxes represent the median with 25–75 percentile range and whiskers represent the 5–95 percentile range. Data points outside of this range are shown individually). **f** Endogenous Treacle localization 2 h after I-Ppo1 transfection in control siRNA and TOPBP1 siRNA transfected cells (percentage of cells with > 2 Treacle caps are indicated; one of two experiments is shown). **g** TOPBP1 localization in Treacle-depleted U2OS cells and control cells 2 h after I-Ppo1 transfection (percentage of cells with > 2 TOPBP1 caps are indicated; one of two experiments is shown). **h** HA-immunoprecipitations from 293FT cells transfected with the indicated HA-tagged Treacle variants, in the presence or absence of ATMi and ATRi and with and without IR treatment as indicated. STTT mutated Treacle: S171A, T173A, T203A, T210A. All scalebars = 10 μm. Source data are provided as a Source Data file.

Downregulation of TOPBP1 expression in U2OS cells by siRNA significantly compromised transcriptional inhibition in response to I-Ppo1 expression (Fig. 4e and Supplementary Fig. 8a). Moreover, downregulation of TOPBP1 also blocked nucleolar segregation in the majority of cells (Fig. 4f). Based on the observation that TOPBP1 interacts with components of the MRN complex[29,30], we considered the possibility that the MRN complex may recruit TOPBP1 in the nucleoli through direct interaction. Consistent with this idea, downregulation of Treacle, the nucleolar adaptor for NBS1, also severely compromised TOPBP1 nucleolar recruitment and cap formation in response to rDNA damage (Fig. 4g). However, co-immunoprecipitation experiments revealed that HA-tagged Treacle mutants that lack the NBS1 interaction sites[21] still efficiently co-immunoprecipitated TOPBP1. Moreover, TOPBP1–Treacle complex formation was constitutive and did not depend on DNA damage or ATM and ATR activity (Fig. 4h). These data suggested that TOPBP1 may interact with Treacle directly and not via the MRN complex. In support of this idea, and consistent with previous findings[31], we identified Treacle as a candidate TOPBP1-interacting protein by mass spectrometric analysis of TOPBP1-associated proteins both in the presence and absence of DNA damage (Supplementary Fig. 9a, b and Supplementary Data 1).

In summary, these data reveal that TOPBP1 is an important regulator of the nucleolar response to rDNA damage and suggest that the MRN complex and Treacle mediate the recruitment of TOPBP1 in the nucleoli in response to rDNA breaks for subsequent ATR activation.

**Characterization of the TOPBP1–Treacle interaction**. We next sought to characterize the TOPBP1–Treacle interaction in more detail. Treacle consists of three structurally distinct regions: an N-terminal region that contains the phosphorylated Ser and Thr residues required for NBS1 interaction[21], a central region consisting of 10 consecutive acidic and serine-rich sequence stretches and a C-terminal region containing nuclear and nucleolar localization signals (Fig. 5a). In order to map the region of Treacle required for its interaction with TOPBP1, we generated a series of deletion mutants of HA-tagged Treacle and assessed their impact on the interaction with TOPBP1 by co-immunoprecipitation experiments. These data revealed that the main TOPBP1 interaction site is located within the C-terminal part of Treacle and that the NBS1 and TOPBP1 interaction sites on Treacle are distinct and do not overlap (Fig. 5b).

TOPBP1 is a modular protein consisting of nine BRCT domains (numbered from 0 to 8), two of which occur in tandems, one in a unique triple-BRCT configuration, and two appear to function as single domains[32,33]. BRCT 1, 2, 5 and 7 contain the necessary residues for phosphopeptide binding[34]. The AAD is located between BRCT domains 6 and 7 (Fig. 5c). BRCT tandems usually contain a conserved lysine residue that directly contacts the phosphate group of a modified protein ligand[35]. Mutating such residues significantly reduces ligand-binding affinity. We therefore mutated Lys residues in BRCT domains 1, 2, 5 and 7 to Ala in GFP-tagged full-length TOPBP1 and used these constructs in co-immunoprecipitation experiments. The analysis showed that, in accordance with previous work, mutation of TOPBP1 BRCT1 abolished TOPBP1 interaction with RAD9 but not with BLM, whereas mutation of BRCT5 disrupted TOPBP1 interaction with BLM but had little effect on RAD9 binding to TOPBP1 (Fig. 5d)[36,37]. Surprisingly, mutation of three BRCT domains (1, 2 and 5) completely abolished Treacle binding to TOPBP1, thus suggesting the presence of three phosphorylated TOPBP1 binding sites in Treacle (Fig. 5d). Of the three purified BRCT tandems fused to GST, only BRCT0-2 pulled down significant amounts of Treacle from Hela nuclear extracts (Supplementary Fig. 8c)

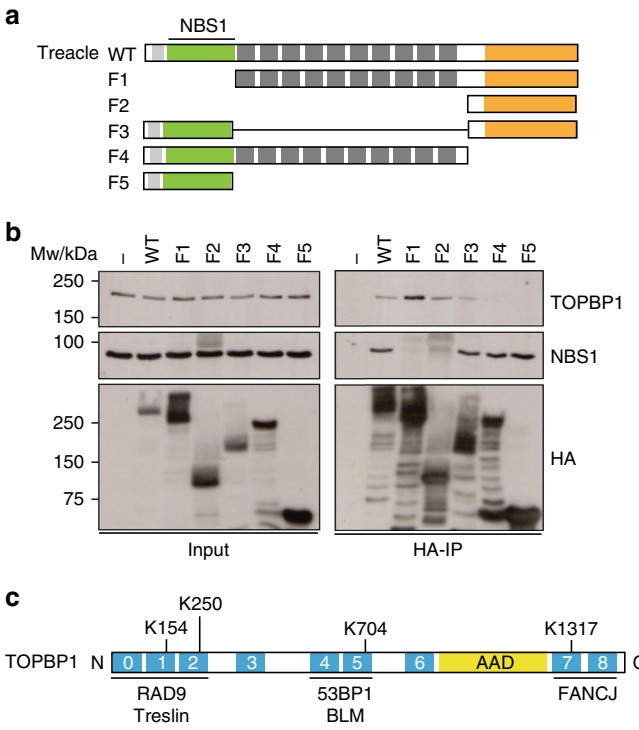

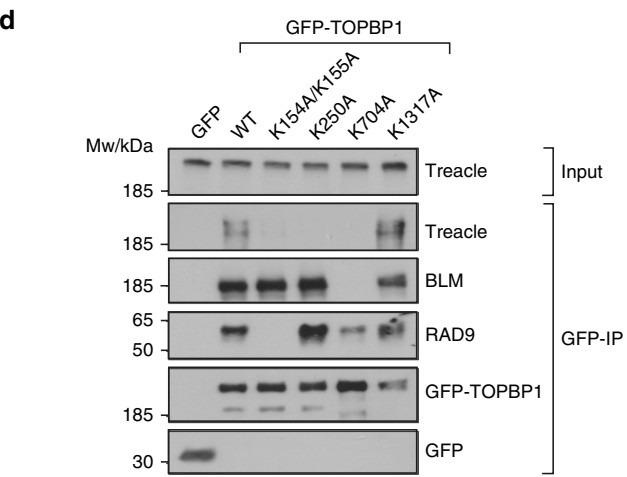

**Fig. 5 Characterization of the TOPBP1–Treacle interaction. a** Schematic showing the layout of conserved domains and motifs in Treacle as well as HA-tagged deletion constructs of Treacle, lacking either the N-terminal region (F1), both the N-terminal and the central repeat region (F2), the central repeat region (F3), the C-terminal region (F4) or both the central repeat and the C-terminal region (F5). HA-immunoprecipitation of full-length and five deletion constructs depicted in **a**. **c** Schematic showing the layout of conserved domains and motifs in TOPBP1. Names of known TOPBP1-binding partners are shown below the domains they interact with and phospho-interacting Lys residues in the three BRCT tandems are highlighted. AAD: ATR-activating domain. **d** GFP-pulldowns from 293FT cells transfected with the indicated GFP-tagged TOPBP1 constructs. Source data are provided as a Source Data file.

suggesting that interaction of Treacle with the isolated TOPBP1 BRCT4 + 5 may be too weak for detection in a GST-pulldown experiment.

**A conserved TOPBP1 phospho-interaction motif in Treacle.** Since complex formation with Treacle requires the phosphopeptide-binding function of three TOPBP1 BRCT domains, we next

mapped phosphorylation sites within the C-terminal region of Treacle by mass spectrometry. Several Ser/Thr phospho-residues were identified with high (>90%) positional probability (Supplementary Data 2). Of these, only a few are conserved. Three phosphorylation sites (Thr1223, Ser1227 and Ser1228) within a highly conserved acidic sequence stretch stand out because along with their surrounding amino acids, they bear significant similarities to known TOPBP1 BRCT0-2 interaction motifs in RAD9 and Treslin (Supplementary Fig. 8d)[31,36]. Interestingly, a highly conserved Ser/Gln (SQ) motif (the ATM/ATR target motif) previously suggested to interact with NBS1[22] is located immediately downstream of the putative BRCT0-2 interaction site. In addition, amino acids N-terminal of this SQ motif closely resemble the TOPBP1 BRCT4-5 interaction site in BLM (Supplementary Fig. 9d)[37,38]. Mutating the central two Ser residues within the putative TOPBP1 BRCT0-2 recognition site severely compromised TOPBP1–Treacle interaction (Fig. 6a). Mutation of the conserved SQ motif also slightly reduced the binding of HA-tagged Treacle to endogenous TOPBP1.

We next sought to investigate if mutation of these phospho-residues in Treacle would affect TOPBP1 recruitment in the nucleoli and transcriptional repression in response to rDNA breaks. To this end, we generated cell lines stably expressing siRNA-resistant GFP-tagged full-length Treacle (wild type and point mutants, respectively) and assessed in each cell line how well ectopic expression of Treacle could rescue the effect of endogenous Treacle downregulation. The analysis showed that in mock-transfected cells, downregulation of endogenous Treacle almost completely abrogated TOPBP1 nucleolar accumulation (Fig. 6b, c). This defect was rescued to control levels by stable ectopic expression of GFP-tagged wild-type Treacle. However, none of the mutants was able to fully complement for the loss of endogenous Treacle expression. Notably, the triple mutant was almost completely defective for TOPBP1 accumulation. In addition, the mutant Treacle-expressing cells showed significantly higher levels of nucleolar EU incorporation after I-Ppo1 transfection when endogenous Treacle was depleted, suggesting a defect in I-Ppo1-induced transcriptional repression in the absence of Treacle–TOPBP1 interaction (Fig. 6d).

In summary, these data reveal that phosphorylation-dependent TOPBP1 binding to a conserved sequence motif within the Treacle C-terminal domain is essential for its recruitment in the nucleoli and is necessary for optimal transcriptional repression in response to rDNA breaks.

**Treacle- and TOPBP1-mediated nucleolar recruitment of ATR.** The mechanism of ATR activation by TOPBP1 is not well understood. We thus considered the possibility that TOPBP1 may mediate ATR recruitment in the nucleoli in response to rDNA breaks. ATR localization in the nucleoli was strongly reduced in Treacle- and TOPBP1-depleted cells (Fig. 7a, b), suggesting that Treacle- and TOPBP1-mediated accumulation of ATR in the nucleoli may be required to ensure transcriptional repression and nucleolar segregation in response to rDNA breaks. In support of this idea, we observed that TOPBP1 overexpression not only triggered nucleolar segregation as previously reported[28], but it also mediated robust ATR accumulation in nucleolar caps (Fig. 7d). ATR activation by TOPBP1 relies on a conserved tryptophan residue that lies within the AAD (W1145). Both nucleolar segregation and ATR accumulation were completely abrogated upon overexpression of a W1145R mutant version of TOPBP1 even though this mutant was still enriched in the nucleoli where it co-localized with Treacle (Fig. 7d, Supplementary Fig. 8d). In Treacle-depleted cells, overexpression of

wild-type TOPBP1 did not trigger nucleolar segregation (Fig. 7e). In summary, these findings support the idea that the nucleolar response to rDNA breaks depends on Treacle- and TOPBP1-mediated accumulation of ATR in the nucleoli.

**Loss of Treacle and TOPBP1 impairs rDNA repair.** We next aimed to establish the physiological significance of TOPBP1/ATR-mediated transcriptional inhibition and nucleolar segregation. Previous studies suggested that these events are required for efficient repair of rDNA breaks[17–19]. We therefore studied recruitment of the DSB repair machinery in response to rDNA breaks in Treacle- and TOPBP1-deficient cells. Depletion of either Treacle or TOPBP1 by siRNA prior to rDNA damage resulted in significantly reduced H2AX phosphorylation and 53BP1 accumulation in the nucleolar periphery. Moreover, Treacle- and TOPBP1-depleted cells were also unable to efficiently recruit RPA and the homologous recombination (HR) factors BRCA1 and RAD51 to nucleolar caps after I-Ppo1 treatment (Fig. 8a–d).

We next depleted Treacle and TOPBP1 by siRNA and measured cell viability in cells transfected with I-Ppo1 mRNA. Depletion of both Treacle and TOPBP1 resulted in a reduced ability of cells to survive DNA damage induced by I-Ppo1 transfection (Fig. 8e). In addition, NBS1ΔN cells were also hypersensitive to I-Ppo1 mRNA transfection when compared to wild-type U2OS cells (Fig. 8f). While the reduced survival may be caused by defective rDNA repair, we cannot rule out a contribution of defective DSB repair at I-Ppo1 sites that are located outside of the rDNA repeats.

Taken together, these data highlight the importance of Treacle as a nucleolar-specific adaptor for NBS1 and TOPBP1 and support a model in which these DDR factors accumulate in the nucleoli upon rDNA breakage in order to recruit and activate the ATR kinase, whose activity is required to repress Pol I transcription and mediate migration of the broken rDNA repeats out of the nucleoli into nucleolar caps, where they can be efficiently accessed by the DSB repair machinery (Fig. 8g).

**Discussion**
The findings reported here reveal a role of ATR kinase activity in the suppression of Pol I transcription and nucleolar segregation in response to rDNA breaks. ATR's role in DSB-induced cell-cycle checkpoint activation is well established[39]. However, its role in the cellular response to DSBs was mostly assigned to S/G2 phases of the cell cycle when sister chromatids are present. This is because extensive regions of RPA-coated ssDNA (RPA-ssDNA), a well-established signal for ATR activation, are not generated at sites of DSBs in G1-phase cells as DNA end resection that generates 3′ ssDNA overhangs is suppressed in G1 (ref. [40]). Despite this well-established model, evidence for ATR activation in G1-phase cells has been recently presented[41,42]. We report here that repression of Pol I transcription and nucleolar segregation are dependent on ATR activity throughout the cell cycle, even in G1 phase, when no sister chromatids are present, and resection of broken rDNA does not take place. Our data thus challenge the current view that RPA-ssDNA is an essential signal for ATR activation in response to DSBs and suggest that ATR may be efficiently activated without RPA-ssDNA in the nucleoli. Accordingly, there is evidence that extensive regions of RPA-ssDNA may not be the signal for ATR activation in response to DSBs also outside of the nucleoli[43–45].

An important element in the process of transcriptional repression and nucleolar segregation in response to rDNA breaks is the accumulation of the MRN complex and TOPBP1 in the nucleoli. Depletion of Treacle not only abrogates recruitment of both MRN and TOPBP1 in the nucleoli, but also results in

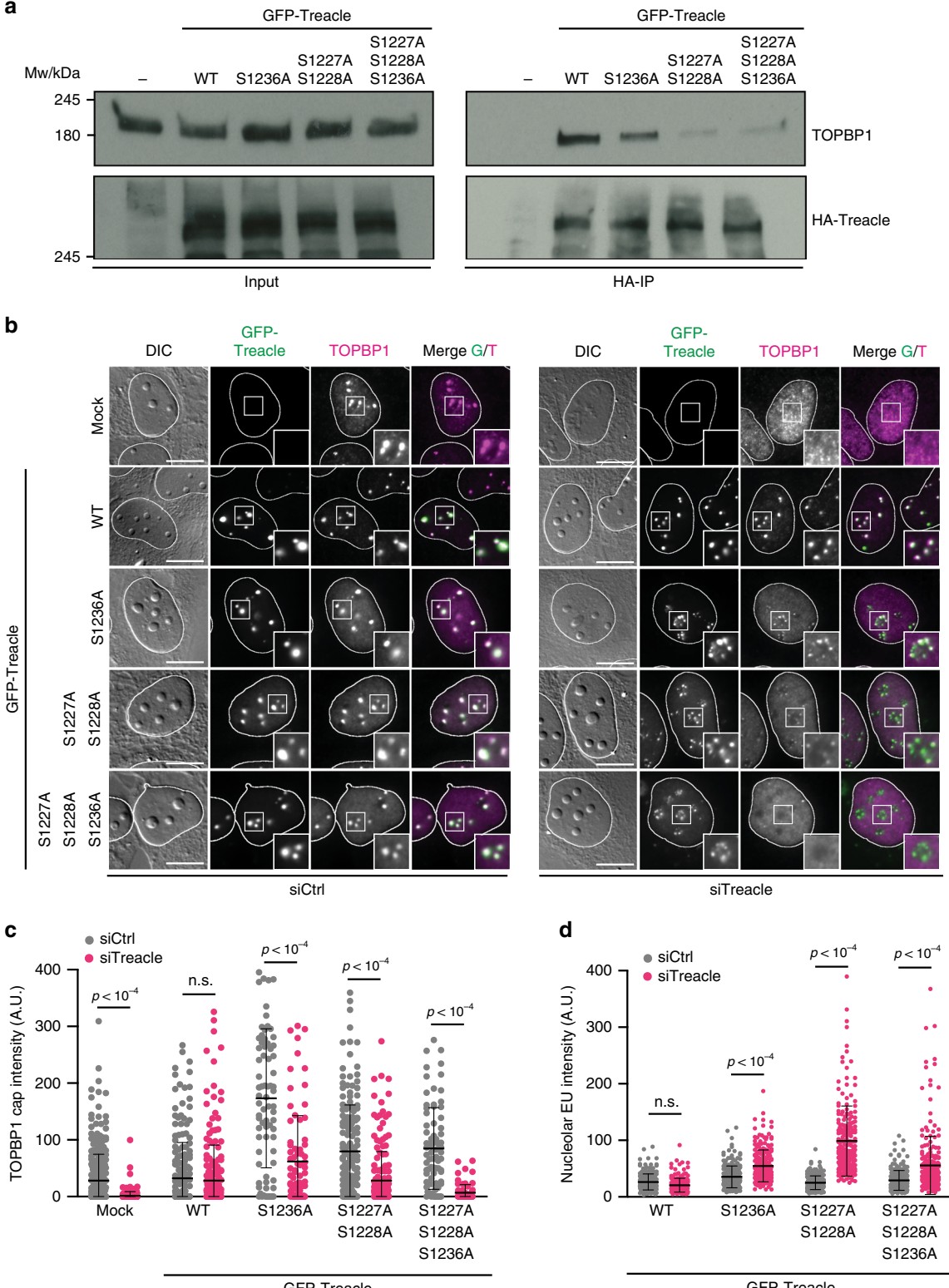

defective recruitment of ATR in the nucleoli and renders cells incapable to undergo nucleolar segregation. However, Treacle-mediated recruitment of the MRN complex and TOPBP1 are unlikely to represent the initial signal for ATM and ATR activation in the nucleoli. We therefore consider it more likely that Treacle lies in the center of an auto-amplification mechanism, which ultimately leads to the accumulation of ATR in the nucleoli. Several lines of evidence support this notion: first,

efficient Treacle-mediated accumulation of NBS1 and TOPBP1 requires ATM and ATR activity, indicating that the initial activation of these kinases at sites of rDNA breaks occurs independently of Treacle. Second, TOPBP1 recruitment in the nucleoli occurs independently of ATR in a subset of cells, and third, overexpression of TOPBP1 leads to ATR-dependent transcriptional repression in the nucleoli followed by nucleolar segregation even in the absence of rDNA breaks[28].

**Fig. 6 A conserved TOPBP1 phospho-interaction motif in Treacle. a** HA-immunoprecipitations from 293FT cells transfected with the indicated HA-tagged Treacle variants. **b** TOPBP1 localization 2 h after I-Ppo1 transfection in GFP-Treacle-expressing cell lines after treatment with control siRNA (left panel) and Treacle siRNA (right panel). **c** Quantification of the experiment in **b**, mock: siCtrl ($n = 301$) versus siTreacle ($n = 290$), WT: siCtrl ($n = 226$) versus siTreacle ($n = 242$), S1236A: siCtrl ($n = 75$) versus siTreacle ($n = 79$), S1227A/S1228A: siCtrl ($n = 169$) versus siTreacle ($n = 177$) and S1227A/S1228A/ S1236A: siCtrl ($n = 77$) versus siTreacle ($n = 69$) cells. **d** Quantification of nucleolar EU incorporation after I-Ppo1 transfection in GFP-Treacle-expressing cell lines after treatment with control siRNA and Treacle siRNA. WT: siCtrl ($n = 216$) versus siTreacle ($n = 200$), S1236A: siCtrl ($n = 224$) versus siTreacle ($n = 224$), S1227A/S1228A: siCtrl ($n = 222$) versus siTreacle ($n = 227$) and S1227A/S1228A/S1236A: siCtrl ($n = 229$) versus siTreacle ($n = 232$) cells. **c**, **d** Statistical significance was calculated using a one-way analysis of variance and Sidak's multiple comparison test. Bars represent mean ± S.D. All scale bars = 10 μm. Source data are provided as a Source Data file.

Our data show that the interaction between Treacle and TOPBP1 is constitutive and DNA damage independent. The TOPBP1 interaction site in Treacle closely resembles known interaction motifs of the TOPBP1 BRCT0-2 module in RAD9 and Treslin[31,36,46]. In addition, our data are consistent with a constitutive and direct interaction between Treacle and TOPBP1 BRCT0-2. While BRCT4 + 5 appear to be also required for stable association between TOPBP1 and Treacle in cells, they do not efficiently interact with Treacle on their own. The reason for this is currently not clear but the sequence similarity of the motif around Ser1236 in Treacle with the TOPBP1 interaction site in BLM makes it a promising candidate for a TOPBP1 BRCT4 + 5 interaction site[37]. Further biochemical work is needed to identify the TOPBP1 BRCT4 + 5 interaction site in Treacle.

Our analysis revealed that DNA end resection of broken rDNA repeats takes place in S/G2 cells. This observation differs from the data presented by a previous work[16] showing cell-cycle independent end resection of the rDNA breaks. However, the discrepancy in the results between the two studies could be attributed to the different timing of the resection assay. We observed resection activity at early time points, and in particular within the first 2 h after transfection of the I-Ppo1 mRNA. At later time points (4 h), we also started to detect resection activity in G1-phase cells, especially in untransformed RPE-1 cells (Supplementary Fig. 7a, b), consistent with the data published by van Sluis and McStay, who assessed resection exclusively at late time points (6 h) after I-Ppo1 mRNA transfection[16]. We therefore propose that prolonged exposure of cells to I-Ppo1 generates persistent DSBs, which may be resected also in G1-phase cells.

Using live-cell imaging, we observed rapid fusion of FC and DFC in the nucleoli upon I-Ppo1 expression throughout the cell cycle, followed by "floating" of the fused FC and DFC to the nucleolar surface where they remain anchored in the nucleolar periphery. Moreover, we found that fusion of FC/DFC and nucleolar segregation occurs downstream of ATR activation, but independently of resection. This is remarkably different from what was observed for DSBs induced outside of the nucleoli, which are resected in an ATM-dependent, but ATR-independent manner[4,5].

Nucleolar material exhibits liquid-demixing behavior[47]. Thereby, FC, DFC and GC within the nucleoli may represent coexisting immiscible liquid phases determined by differences in the biophysical properties of their constituents[48]. Ongoing transcription is an essential element of the maintenance of nucleolar structure because it may prevent coalescence of DFC-surrounded GCs to form larger centers[20]. Thus, when transcription is inhibited, GC and DFC start to fuse, which may induce a liquid-phase de-mixing process that results in the "floating" of the exposed GC to the nucleolar periphery. Our live-cell imaging data clearly support such a liquid-like behavior of the nucleoli and suggests liquid de-mixing processes as the driving force behind nucleolar segregation. In summary, we propose that nucleolar reorganization induced by rDNA breaks may be a consequence of inhibition of transcription resulting in enhanced ability of FC and DFC to coalesce.

The mechanism of Pol I inhibition in response to rDNA breaks is not yet clear. ATM-dependent inhibition of RNA polymerase II transcription in response to DSBs has been shown to involve local histone H2A ubiquitylation and the recruitment of a specific chromatin remodeler as well as the cohesion complex[10,11,14]. Posttranslational histone modifications also seem to be key events in Pol I inhibition because ATM-dependent H2B S14 phosphorylation by MST2 kinase is required for transcriptional repression in the nucleoli[19]. Direct modification of the Pol I transcription machinery by ATM and ATR may also be involved as several key Pol I transcription factors appear to be targeted by PIKKs[49]. However, the fact that transcriptional inhibition is dependent on both PIKK activity as well as downstream signaling events involving CHK1 and CHK2 kinases, it is likely that more than one effector molecule mediates DSB-induced transcriptional inhibition in the nucleoli.

## Methods

**Cell culture and drug treatments**. All cell lines were grown in a sterile cell culture environment and routinely tested for mycoplasma contamination. 293FT (transformed human embryonal kidney cell line), U2OS (human osteosarcoma cell line), HeLa (cervical carcinoma cell line) and RPE-1 (immortalized human retinal pigmented cell line) cells were cultured in Dulbecco's modified Eagle medium (DMEM), supplemented with 10% fetal calf serum (FCS), 2 mM L-glutamine and penicillin–streptomycin antibiotics under standard cell culture conditions in a CO$_2$ incubator (37 °C; 5% CO$_2$). U2OS cell lines stably transfected with GFP-Treacle and tetracyclin repressor expression constructs were cultured in the presence of 200 μg/ml Zeocin and 10 μg/ml Blasticidin S (Invitrogen). U2OS cell lines stably transfected with NBS1-mNG expression constructs and with GFP-ATR expression constructs (gift from J. Lukas) were cultured in the presence of 1 μg/ml Puromycin. For pulsed EU incorporation, cells were incubated for 30 min in medium containing 1 mM EU. The Click-iT EU Alexa Fluor 594 Imaging Kit (Thermo Fisher Scientific) was used for EU detection. Unless stated otherwise, the following compounds were used at the indicated final concentrations: ATM inhibitor KU-55933 (5 μM; Selleckchem), ATR inhibitor VE-821 (5 μM; Selleckchem), ATR inhibitor HY-19323 (5 μM; MedChemExpress), CHK1/CHK2 inhibitor AZD7762 (300 nM; TargetMol), CHK1 inhibitor GDC-0575 (300 nM; MedChemExpress). X-ray irradiation of cells was performed with a YXLON.SMART 160E-1.5 device (150 kV, 6 mA; YXLON International SA) delivering 11.8 mGy per second. Soft X-rays were largely filtered out with a 3-mm aluminum filter.

**Cloning and mutagenesis**. The pcDNA4-TO-strep-HA-GFP-Treacle construct for GFP-Treacle expression, the deletion mutants F1-F5 and the point mutant S171A/ T173A/T203A/T210A (STTT) has been described[21]. Mutant derivatives S1236A, S1227A/S1228A and S1227A/S1228A/S1236A of this construct were generated by site-directed mutagenesis using the QuikChange II Site-Directed Mutagenesis kit (Agilent Technologies). Sequences of the mutagenesis primers were:
  S1236A forward: GGAGAGACTGGGCTGGCGCCACATC
  S1236A reverse: GATGTGGTGGCGCCAGCCCAGTCTCTCC
  S1227A/S1228A forward:
GAGACCGCAGCAGCAGAGGCCGCCGAGGATGATGTGGTG
  S1227A/S1228A reverse:
CACCACATCATCCTCGGCGGCCTCTGCTGCGGTCTC
  pcDNA5/FRT/TO-NBS1-mNG were generated by PCR amplification of mNG from pmNeonGreen-C1 (Allele Biotechnology) and ligation into pcDNA5/FRT/ TO-NBS1-WT, -R28A, -K160M and R28A/K160M, respectively[50]. Plasmids pIRES V5 I-Ppo1 and pIRES I V5 -Ppo1(H98A) for I-Ppo1 mRNA purification (gift from Brian McStay) were described[16]. pIRESneo2-TOPBP1-WT, -K154A/K155A (BRCT1), -K704A (BRCT5), -K1317A (BRCT7) were described[37,51,52]. The mutant derivative W1145R of this construct was generated by site-directed mutagenensis using the QuikChange II Site-Directed Mutagenesis kit (Agilent Technologies). Sequences of the mutagenesis primers were:

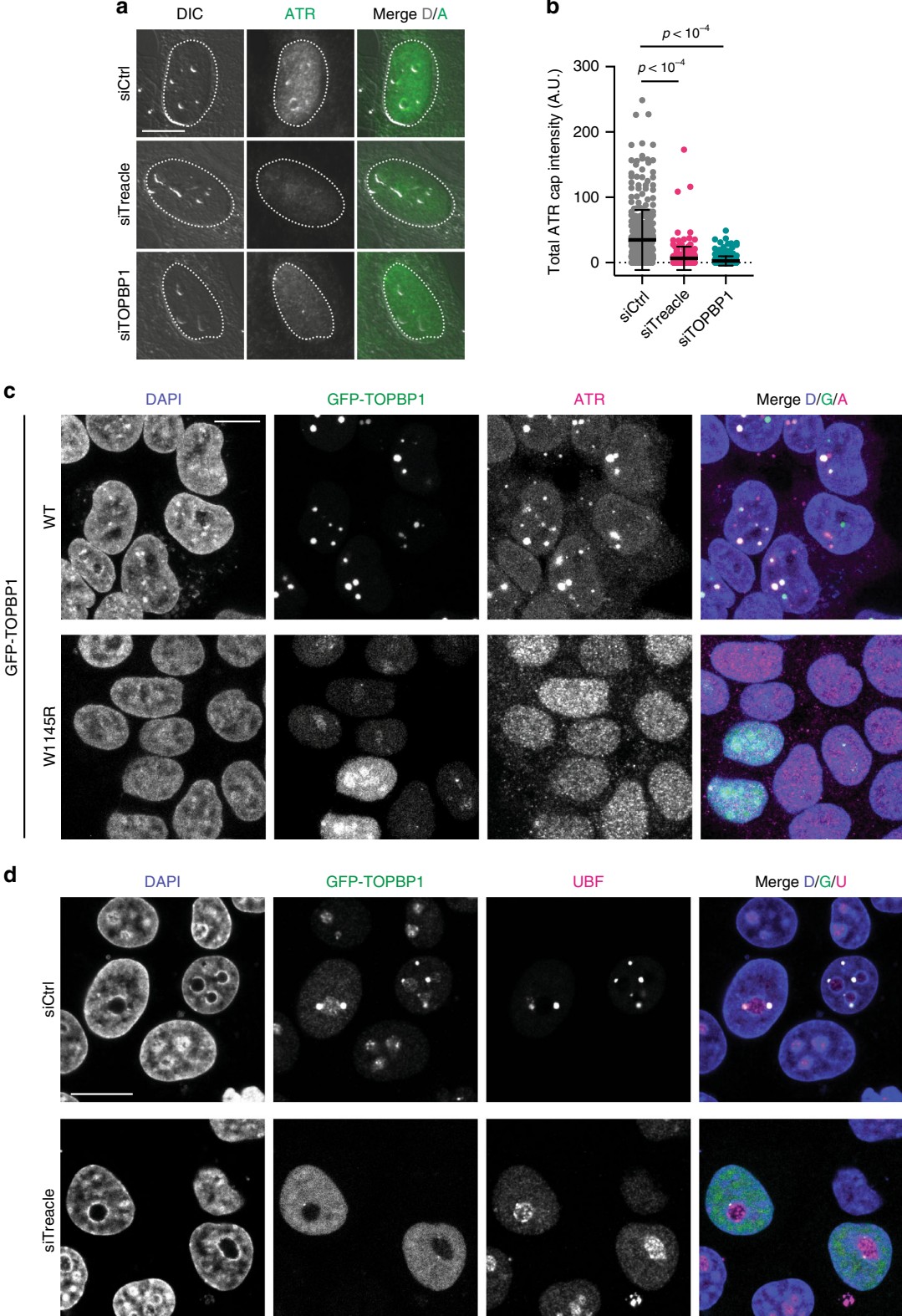

**Fig. 7 Treacle- and TOPBP1-mediated nucleolar recruitment of ATR. a** ATR localization in Treacle- and TOPBP1-depleted U2OS cells and control cells 2 h after I-Ppo1 transfection. **b** Quantification of the experiment in **a**, siCtrl ($n = 364$), siTreacle ($n = 214$) and siTOPBP1 ($n = 245$) treated cells. Statistical significance was calculated using a one-way analysis of variance and Sidak's multiple comparison test. Bars represent mean ± S.D. **c** ATR localization in 293-T cells overexpressing GFP-TOPBP1 wild type and W1145R AAD point mutant. **d** UBF localization in Hela cells overexpressing GFP-TOPBP1 and transfected with control siRNA and Treacle siRNA, respectively. All scale bars = 10 μm. Source data are provided as a Source Data file.

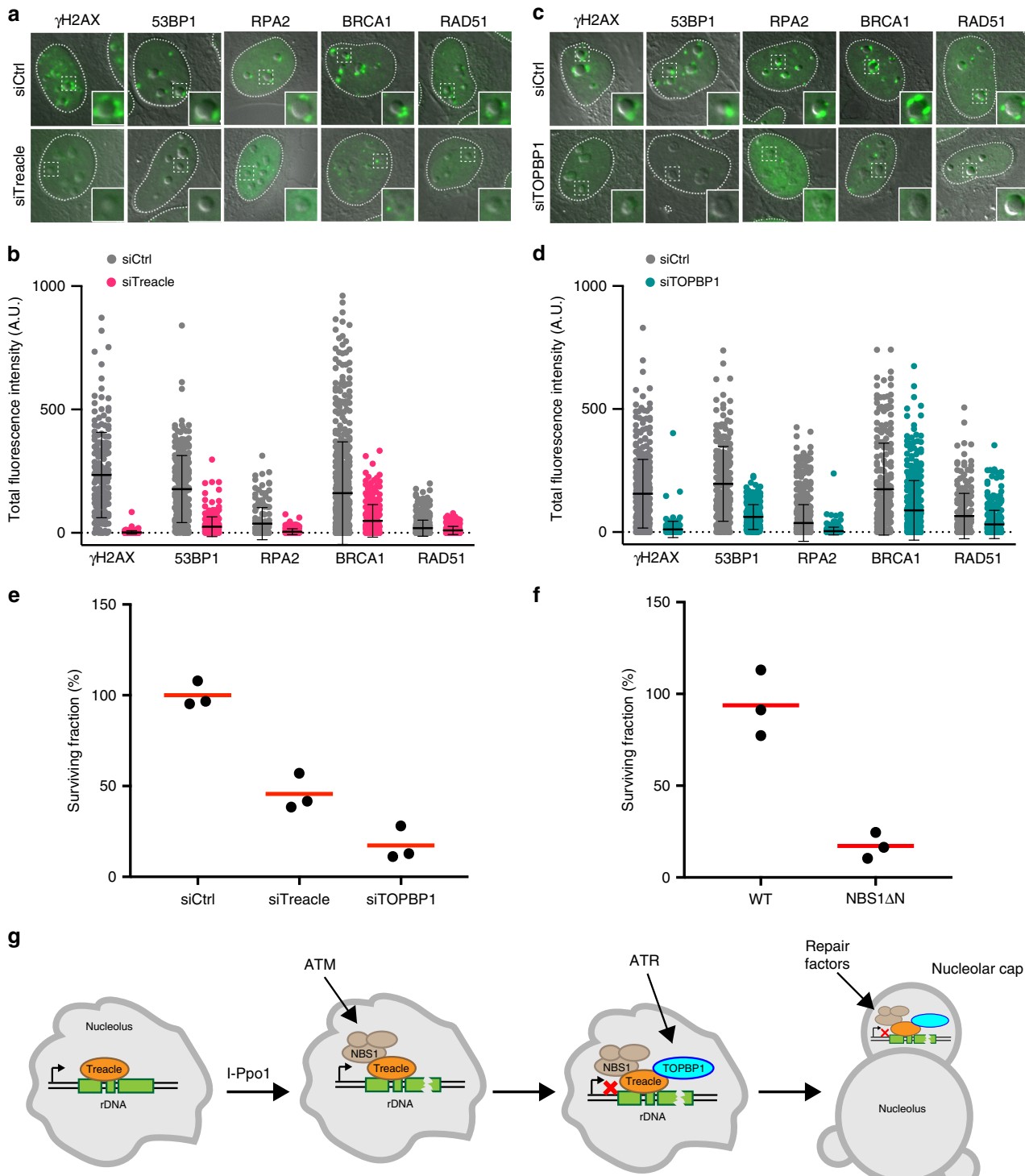

**Fig. 8 Loss of Treacle and TOPBP1 impairs rDNA repair. a** γH2AX, 53BP1, RPA2, RAD51 and BRCA1 localization in control depleted (siCtrl) and Treacle-depleted (siTreacle) U2OS cells 2 h after I-Ppo1 transfection. **b** Quantification of the experiment in **a**, γH2AX: siCtrl (n = 177) versus siTreacle (n = 200), 53BP1: siCtrl (n = 261) versus siTreacle (n = 239), RPA2: siCtrl (n = 152) versus siTreacle (n = 168) BRCA1: siCtrl (n = 525) versus siTreacle (n = 246) and RAD51: siCtrl (n = 525) versus siTreacle (n = 246) cells (all datapoints are shown, bar represents mean, error bars represent S.D.) **c** γH2AX, 53BP1, RPA2, RAD51 and BRCA1 localization in control depleted (siCtrl) and TOPBP1-depleted (siTOPBP1) U2OS cells 2 h after I-Ppo1 transfection.
**d** Quantification of the experiment in **c**, γH2AX: siCtrl (n = 403) versus siTOPBP1 (n = 205), 53BP1: siCtrl (n = 239) versus siTOPBP1 (n = 256), RPA2: siCtrl (n = 649) versus siTOPBP1 (n = 359) BRCA1: siCtrl (n = 228) versus siTOPBP1 (n = 348) and RAD51: siCtrl (n = 228) versus siTOPBP1 (n = 348) cells (all datapoints are shown, bar represents mean, error bars represent S.D.). **e** Clonogenic survival analysis of I-Ppo1-transfected control (siCtrl), Treacle-depleted (siTreacle) and TOPBP1-depleted (siTOPBP1) U2OS cells (red bars represent the mean). **f** Clonogenic survival analysis of I-Ppo1-transfected WT and NBS1DN U2OS cells (red bars represent the mean). **g** Model (see text for details). Source data are provided as a Source Data file.

W1145R forward:
GCCTTCCCAAAATGAACAGATCATTCGGGATGACCCTAC
W1145R reverse:
GTAGGGTCATCCCGAATGATCTGTTCATTTTGGGAAGGC.

**Generation of stable cell lines**. To isolate cell lines with low GFP-Treacle expression, U2OS cells were co-transfected with pcDNA6/TR (Invitrogen) and pcDNA4-TO-strep-HA-GFP-Treacle[21] constructs and stably transfected cells were selected in the presence of 400 μg/ml Zeocin and 5 μg/ml Blasticidin S (Invitrogen). Subsequently, clones with low levels of Doxycycline-independent "leaky" expression of GFP-Treacle were isolated. NBS1ΔN cells stably expressing NBS1-mNG were generated by co-transfection of pcDNA5/FRT/TO-NBS1-mNG with pBabe-Puro, followed by selection with 1 μg/ml Puromycin. To isolate cell lines with homogenous NBS1-mNG expressing, pooled colonies were sorted using a BD FACSAria II 4L cell sorter.

**Generation and characterization of the NBS1ΔN cell line**. For the generation of NBS1 knock-out cells, the sgRNA target sequence (50-GCGTTGAGTACGTTGT TGGA-30) was cloned into the PX330-U6-hSpCas9 vector and verified by sequencing. The targeting vector was then transfected into U2OS cells, and clonal cell lines were isolated by dilution in 96-well plates. Single clones were analyzed for NBS1 expression by immunofluorescence[50]. Several clones without detectable NBS1 expression were isolated and one of them was further characterized. Frame-shift-inducing insertion/deletion (indel) mutations in all of the three NBS1 alleles present in U2OS cells were verified by Sanger sequencing of a PCR-amplified genomic fragment that was cloned in the pCR II Blunt TOPO plasmid.

**RNA transfection**. The control siRNA (siCtrl) and siRNA against Treacle (siTreacle), TOPBP1 (siTOPBP1), ATR (siATR) and CtIP (siCtIP) were obtained from Microsynth AG. The sequences of the siRNAs were: siCtrl: UGGUUUACA UGUCGACUAA-dTdT, siTreacle: CCACCAUGGGUUGGAACUAAAUU-dTdT, siTOPBP1: ACAAAUACAUGGCUGGUUA-dTdT, siATR: CCUCCGUGAUGUU GCUUA-dTdT and siCtIP: GCTAAAACAGGAACGAATC-dTdT. Cells were grown in six-well plate 24 h prior the transfection. The siRNA transfection was done with 10 pM siRNA using Lipofectamine RNAiMAX (Invitrogen). Cells were split on coverslips 24 h after the transfection and harvested 72 h after the transfection.

For the generation of the in vitro transcribed I-Ppo1, we used the vectors pIRES I-Ppo1 wild type and H98A (a gift from Brian McStay[16]). Plasmids were linearized with NotI and transcribed in vitro using the MEGAscript T7 kit (Ambion) according to the manufacturer's instructions. The I-Ppo1 mRNA was then polyadenylated using the Poly(A) tailing kit (Ambin) according to the manufacturer's instructions. For I-Ppo1, mRNA transfection cells were grown on glass coverslips in six- or 24-well plate 24 h prior the I-Ppo1 transfection. The mRNA transfection was done using the Lipofectamine MessengerMax Reagent (Invitrogen) according to the manufacturer's instruction.

**SDS-PAGE and western blotting**. SDS-PAGE and western blotting were performed using BioRad 4–15% Mini-PROTEAN TGXTM Precast Gels. Additional SDS-PAGE and western blotting was done using Bolt 4–12% Bis-Tris Plus gels from Thermo Fisher Scientific. The following antibodies were used at the indicated dilutions: BLM (goat, Abcam, ab5446, 1/500), GFP (mouse, Roche, 11814460001, 1/5000), HA (mouse, Santa Cruz Biotechnology, sc-57592, 1/1000), NBS1 (rabbit, Abcam, ab32074, 1/2000), NBS1 (mouse, GeneTex, GTX70224, 1/1000), TOPBP1 (A300-111A, Bethyl Laboratories, 1/5000), TOPBP1 (rabbit, Abcam, Ab2402, 1/1500), ATM pS1981 (rabbit, Epitomics, 2152-1, 1/5000), ATM (rabbit, Calbiochem, PC-116, 2/250), Treacle (rabbit, Sigma, HPA038237, 1/500), RAD9 (rabbit, Abcam, ab70810, 1/3000), RAD50 (mouse, GeneTex, 13B3, GTX70288, 1/1000), MRE11 (mouse, Abcam, 12D7, ab214, 1/500), Tubulin (mouse, Sigma, DM1A; T6199, 1/2000), RPA-pS4/S8 (rabbit, Bethyl laboratories, A700-009, 1/1000), RPA2 (mouse, Abcam, ab2175, 1/10000), CHK1 pS317 (rabbit, Cell Signaling, 2344, 1/500), CHK1 pS345 (rabbit, Cell Signaling, 2348, 1/500) CHK1 (mouse, Santa Cruz Biotechnology, sc-8408, 1/1000), CHK2 pT68 (rabbit, Cell Signaling, 2661, 1/1000), CHK2 (rabbit, Cell Signaling, 2662, 1/1000).

**Immunoprecipitations**. For GFP-Trap immunoprecipitations, cells were washed twice in phosphate-buffered saline (PBS), and lysed in IP buffer (100 mM NaCl, 0.2% Igepal CA-630, 1 mM $MgCl_2$, 10% glycerol, 5 mM NaF, 50 mM Tris-HCl, pH 7.5), supplemented with cOmplete EDTA-free protease inhibitor cocktail and 25 U/ml Benzonase (Novagen). After nuclease digestion, NaCl and EDTA concentrations were adjusted to 200 mM and 2 mM, respectively, and lysates were cleared by centrifugation. Lysates were then incubated with 15 μl of GFP-Trap magnetic agarose beads (ChromoTek) with end-to-end mixing at 4 °C. Immunoglobulin–antigen complexes were washed five times with IP buffer before elution in 2× SDS sample buffer. For HA immunoprecipitations, cells were washed twice in PBS, and lysed in IP buffer (100 mM NaCl, 1 mM $MgCl_2$, 2 mM EDTA, 10% glycerol, 5 mM NaF, 0.2% Nonidet 40 (IP40), 50 mM Tris-HCl, pH 7.6), supplemented with cOmplete EDTA-free protease inhibitor cocktail and 25 U/ml Benzonase (Novagen). After nuclease digestion, lysates were cleared by centrifugation. Lysates were then incubated with 40 μl of anti HA-Agarose beads

(Sigma) with end-to-end mixing at 4 °C. Immunoglobulin–antigen complexes were washed three times with IP buffer before elution in 2× SDS sample buffer.

**GST pulldowns**. GST fusion proteins were expressed in the competent *Escherichia coli* expression strain BL21(DE3)pLysS (Promega) followed by protein purification with glutathione–Sepharose beads. A total of 5 μg of purified GST-TOPBP1 BRCT fusion proteins were mixed with 33 μl of HeLa nuclear extract (7 mg/ml; Ipracell) and incubated for 1.5 h at 4 °C. Glutathione–Sepharose beads (GE Healthcare) were added and the suspension was incubated at 4 °C for 1.5 h on a rotating wheel. The beads were washed three times with wash buffer (50 mM Tris, pH 7.5, 150 mM NaCl, 1 mM DTT, and 0.2% NP-40) and bound proteins were eluted by addition of 2× SDS sample buffer.

**Immunofluorescence**. Cells were grown on glass coverslips and fixed with either ice-cold methanol for 10 min or with 4% buffered formaldehyde for 15 min at room temperature, and subsequently permeabilized for 5 min in PBS containing 0.3% Triton X-100. Following 1 h of blocking in blocking buffer (10% FBS, 3% BSA in PBS), primary antibody incubations were performed at 4 °C overnight. Coverslips were washed three times with PBS and secondary antibody incubations were performed for 1 h at room temperature in the dark. After washing with PBS for three times, coverslips were mounted on glass microscopy slides with Vectrashield mounting medium containing 0.5 μg/ml 4′,6-diamidino-2-phenylindole dihydrochloride (DAPI). The following antibodies were used at the indicated dilutions: Treacle (rabbit, Sigma Life Science, HPA038237, 1/100), TOPBP1 (rabbit, Bethyl, A300-111A, 1/500), Nucleophosmin (mouse, Thermofisher, MA5-17141, 1/300), BRCA1 (mouse, sc-6454, Santa Cruz, 1/100), RAD51 (rabbit, Santa Cruz, sc-8349, 1/100), γH2AX (mouse, Millpore, 05-636, 1/500), MDC1 (mouse, Abcam, Ab50003, 1/300), Cyclin A (mouse, BD biosciences, 611269, 1/100), RPA2 (mouse, Abcam, Ab2175, 1/250), RPA2 pS4/S8 (rabbit, Bethyl A300-245, 1/400), 53BP1 (mouse, gift from T. Halazonetis, 1/20) NBS1 (rabbit, Novus, NB100-143, 1/200), MRE11 (mouse, GeneTex, GTX70212, 1/1000), UBF (mouse, Santa Cruz, sc-13121, 1/100), ATM (rabbit, Abcam, ab32420, 1/250), ATR (rabbit, Cell Signaling E1S3S, 1/100), V5 (mouse, Abcam, ab27671, 1/500).

**Widefield and confocal microscopy**. Widefieled image acquisition was done on a Zeiss AxioObserver Z1 widefield microscope, equipped with a Lumencor SpectraX illumination system and a Hamamatsu Orca Flash 4.0 V2, sCMOS, cooled fluorescence camera (16 bit, 2048 × 2048 pixel (4 MP), pixel size 6.5 μm). A 63×, 1.4-NA, i-plan apochromat oil-immersion objective was used. For optimal representation in figures, images were adjusted for brightness and exported as RGB TIF files using Fiji[53].

Confocal images were acquired with a Leica SP8 inverse confocal laser scanning microscope with a 63×, 1.4-NA Plan-Apochromat oil-immersion objective. The sequential scanning mode was used and the number of overexposed pixels was kept at a minimum. Fields were recorded at a resolution of 512 × 512 pixels, 8 bit depth or 1024 × 1024 pixels, 8 bit. For optimal representation in figures, maximum intensity projections were calculated and images were adjusted for brightness and exported as RGB TIF files using Fiji[53].

**Image quantification**. The xyz confocal datasets (z-stacks) were analyzed using IMARIS 9.2 (Bitplane). Caps/foci were segmented and counted by the integrated intensity-based spot detection tool and nucleoli were surface rendered to calculate nucleolar volume and sphericity.

Nucleolar caps and DNA damage foci were quantified using CellProfiler 3.0[51]. First, nuclei segmentation was performed by the intensity-based primary object detection module using the DAPI signal. For nucleolar caps and foci segmentation, the primary object detection module was used on the respective channels after applying a feature enhancement filter. Downstream data manipulation and graphical representation of the data were done using R 3.4.2 (R Development Core Team). CellProfiler pipelines and R scripts are available upon request.

Quantification of nucleolar EU incorporation was done with Fiji[53]. Nucleoli were segmented manually by using the DIC channel and a Wacom graphic tablet. Nucleoli were marked as regions of interest (ROI) and added to the ROI manager. Fluorescence intensities were then measured within the ROIs in the EU channel.

**Time-lapse microscopy**. GFP-Treacle- or NBS1-mNG-expressing cells were plated 24 h prior the imaging in the μ-Slide 8-well chamber coverslip (Ibidi). I-Ppo1 transfection was done using Lipofectamine MessengerMax Reagent (Invitrogen) according to the manufacturer's instruction. Cells were followed over time with a Zeiss AxioObserver Z1 widefield microscope equipped with a Definite Focus hardware autofocus control unit. Temperature and $CO_2$ levels were kept at 37 °C and 5% $CO_2$ for the duration of the experiment. A 63×, 1.4-NA, i-plan apochromat oil-immersion objective was used. Phase-contrast images and wavelength detection EGFP were acquired at 3 min intervals over a period of 2–3 h. Movies of individual cells were generated by applying the StackReg plugin in Fiji (Rigid body transformation) to compensate for cell movement. Time-lapse recordings were then animated and exported as GIF files or rendered to mp4 video files with Adobe Photoshop CC 2018.

**Real-time PCR**. Quantification of rRNA transcription by real-time PCR was essentially done as described previously[21]. Unless otherwise indicated, RNA

isolation and cDNA synthesis were carried out 6 h after I-Ppo1 transfection. Real-time PCR was performed using 10 ng of cDNA with SYBR Green I Master (Roche, Cat. no. 04707516001). Samples were analyzed on a LightCycler480 and quantified with LightCycler480 quantification software.

**Clonogenic survival assay after I-Ppo1 transfection**. Cells were treated either with control siRNA (siCtrl), siTreacle or siTOPBP1 and 48 h post transfection, cells were transfected with I-Ppo1 mRNA. To ensure the same conditions, cells were counted and plated in six-well plates 4 h after I-Ppo1 transfection. For each condition, 1000 cells per well were plated, in triplicates. Colonies were grown for 10 days, with medium change every 4 days. Plates were washed twice in PBS and incubated for 1 h in Methylene blue fixation/staining solution (80% EtOH, 1% Methylene blue powder) at room temperature. After rigorous washing with warm tap water, plates were left to dry overnight and colonies were counted manually using an eCount™ colony counter pen.

**Mass spectrometry**. Mass spectrometric experiments were performed on a nanoscale UHPLC system (EASY-nLC1000 from Proxeon Biosystems) connected to an Orbitrap Q-Exactive HF equipped with a nanoelectrospray source (Thermo Fisher Scientific). Peptides were auto-sampled and separated on a 15-cm analytical column (75 μm ID) in-house packed with 1.9 μm C18 beads using a 77-min gradient ranging from 5 to 40% acetonitrile in 0.1% formic acid at a flow rate of 250 nl per min. The mass spectrometer was operated in data-dependent acquisition mode and all samples were analyzed using a previously described "sensitive" acquisition method[52]. All raw data analysis was performed with MaxQuant software suite[54] version 1.5.3.30 supported by the Andromeda search engine[55]. For generation of the theoretical spectral library, a HUMAN.fasta database was extracted from UniProt. Cysteine carbamido-methylation was included as a fixed modification, whereas N-terminal acetylation, methionine oxidation and phosphorylation (S, T and Y), were included as variable modifications. Mass tolerance for precursors was set to 20 ppm in the first MS/MS search and 4.5 ppm in the main MS/MS search. Data were automatically filtered by posterior error probability to achieve a false discovery rate of < 1% (default), at the peptide-spectrum match and the site-specific levels. For enrichment and mass spectrometric analysis of Treacle phosphopeptides, 293T cells were transfected with pcDNA4-TO-strep-HA-GFP-Treacle and collected in high-salt RIPA buffer (50 mM Tris pH7.5, 400 mM NaCl, 1 mM EDTA, 1% Nonidet P-40, 0.1% Na-deoxycholate, 2 mM Na-orthovanadate, 5 mM NaF, 5 mM glycero-2-phosphate, protease inhibitors (Roche)). Proteins from cleared lysate were precipitated in acetone, dissolved in urea, reduced with dithiothreitol, alkylated with chloroacetamide30, and digested using Lys-C and modified sequencing-grade trypsin (Sigma). Protease digestion was terminated by addition of trifluoroacetic acid and peptides were purified using reversed-phase Sep-Pak C18 cartridges (Waters). Phosphopeptides were enriched on a TiO$_2$ column prior to mass spectrometry.

**Statistical analysis**. Graphs were generated with Prism 8 for macOS (GraphPad Software Inc., Version 8.1.2). Unless otherwise indicated, horizontal bars represent the mean values and error bars the standard deviation. In box plot graphs, boxes represent the 25–75 percentile range with median, and whiskers represent the 9–95 percentile range. Data points outside this range are shown individually. Statistical tests were applied as described in the Figure legends and were calculated with Prism 8. In all quantitative immunofluorescence studies, at least 100 random cells per group were scored.

**Reporting summary**. Further information on research design is available in the Nature Research Reporting Summary linked to this article.

## Data availability
The mass spectrometry proteomics data have been deposited to the ProteomeXchange Consortium via the PRIDE[56] partner repository with the dataset identifier PXD015138. The source data underlying Figs. 1d, f–h, 2b, 3c, e, g, 4d, e, h, 5b, d, 6a, c, d, 7b and 8b–d and Supplementary Figs. 1c–e, 3b–d, 7c and 8a are provided as a Source Data file. All other relevant data, CellProfiler pipelines and R scripts are available from the authors upon request.

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

## Acknowledgements

We thank Brian McStay, Jiri Lukas and Thanos Halazonetis for providing valuable reagents, Dafni Pefani for providing technical advice and Dorthe Larsen for sharing unpublished observations. Imaging was performed with equipment maintained by the Center for Microscopy and Image Analysis, University of Zurich. Cell sorting was carried out by the Flow Cytometry Core Facility, University of Zurich. The Blackford lab is supported by a Cancer Research UK Career Development Fellowship (C29215/A20772) to A.N.B. and an MRC WIMM DPhil Studentship (MR/N013468/1) to A.-M.K.S. and A.N.B. The Nielson lab is supported by the Novo Nordisk Foundation Center for Protein Research, the Novo Nordisk Foundation (grant agreement numbers NNF14CC0001 and NNF13OC0006477), and the Danish Council of Independent Research (grant agreement numbers 4002-00051, 4183-00322A and 8020-00220B). The Stucki lab is supported by the Forschungskredit Candoc program of the University of Zurich (FK-18-024) to P.-A.L., a project grant from the Swiss National Foundation (31003A_163141), the Promedica foundation and a grant from the Helmut Horten Foundation.

## Author contributions

The project was conceived and supervised by M.S. Biochemical and cell biological experiments were carried out and data were analyzed by C.M., I.-E.S., A.R., P.-A.L., A.-M.K.S., A.N.B. and M.S. Experimental tools and reagents were generated and characterized by D.Z., C.L., A.Ja., A.Je. and K.K. D.F. supervised D.Z. and A.Je. Mass spectrometry data were contributed by S.J., S.C.L. and M.L.N. The paper was written by M.S. with contributions from other authors.

## Competing interests

The authors declare no competing interests.
