## [Peer Review File · Nature Communications]

Reviewers' comments:

Reviewer #1 (Remarks to the Author):

Stucki and colleagues present new data building on their previous manuscript that Treacle is required for the relocalization of TOPBP1 into the nucleolus and the activation of ATR following the induction of DNA breaks in the rDNA loci. In addition, Stucki and colleagues present evidence to suggest that both ATR and TOPBP1 are required to shut down rDNA transcription and promote nucleolar segregation.

Overall, the data is of a high quality and mostly supports the conclusions made by the authors. The cellular studies assessing the relocalisation of NBS1, Treacle and TOPBP1 to nucleoli and the biochemical studies mapping the phospho-sites on Treacle that bind TOPBP1 have been carried out to a very high standard.

Whilst the data presented does indicate a new function of Treacle in promoting the relocalisation of TOPBP1 into the nucleolus in a phospho-dependent manner, I find the story a little disjointed. It is not clear to me how ATR is activated in response to rDNA DSBs in G1 and G2, how ATR activation relates to rDNA transcriptional inhibition, why ATM and ATR are both required for rDNA transcriptional inhibition and what the relationship is between rDNA DSB repair, rDNA transcriptional inhibition, nucleolar segregation, NBS1 vs TOPBP1 recruitment to nucleoli and G2-associated resection/HR of rDNA DSBs? I appreciate that there are a lot of unanswered questions and the authors should not necessarily be required to answer them all. However, filling some of these gaps will help with the flow of the paper rather than it being a collection of interesting observations.

Specific points:

1. The authors routinely use I-Ppo1 mRNA to induce DSBs in rDNA. However, the authors really need to use a fluorescent marker in their assays to show that they have actually induced comparable levels of DSBs. In addition, since the authors are directly transfecting I-Ppo1 mRNA into cells, they need to demonstrate that in each case that the levels of I-Ppo1 protein are comparable. This is particularly important when using the ATM/ATR inhibitors, given that it is possible these inhibitors are reducing the translation of the I-Ppo1 mRNA.
2. The authors often refer to rDNA DSBs inducing the activation of both ATM and ATR. However, to make this claim the authors really need a fluorescent marker to show that this really is the case. To me, demonstrating a reduction in rDNA transcription when the cells are treated with either ATM or ATR inhibitors is not really good enough. The authors could try using the phospho-ATM and ATR antibodies or ChIPing ATM/ATR to rDNA DSBs.
3. I am a little confused as to why the NBS1DeIN cells do not exhibit an ATM activation defect, when cells containing the commonly found 657Del5 NBS1 mutation does give rise to ATM activation defect.
4. The authors use RPA foci as a marker of rDNA resection. Although this indicates that there is ssDNA, it is not concrete marker of resection. The authors should use the phospho-RPA2 S4/8 antibody and native BrdU foci to provide more solid evidence that these DSBs are being resected. Is the rDNA DSB resection dependent on the nuclease activity of Mre11?
5. Supplementary table 1 should contain all the specific TOPBP1 binding proteins identified by mass spec.
6. The authors make claims that the C-terminal phospho-domain of Treacle, which is necessary for TOPBP1 relocalisation to the nucleoli, is required for nucleolar segregation in response to I-Ppo1 expression. This is not the case. No analysis of nucleolar segregation with the Treacle S>A mutant has been shown.
7. Whilst I appreciate that it is difficult to demonstrate what the outcome of defective rDNA repair is on cell survival, the graphs depicting survival of cells depleted of Treacle, TOPBP1 or NBS1

siRNA following I-Ppo1 mRNA transfection cannot be attributed solely to defective rDNA repair, since the I-Ppo1 also induces DSBs at nuclear loci. A better approach would be to look at chromosome breaks in rDNA using metaphase-FISH, although this will not tell you what outcome a failure to repair rDNA DSBs has on cell survival.

8. Is the AAD of TOPBP1 required to initiate or maintain ATR activation in response to rDNA DSBs? What is ATR being activated by in the nucleoli in G1, when resection does not occur? Have the authors considered looking at R-loops and seeing whether this plays a role in activating ATR?

Minor point:

1. On page 21, line 479, PRE-1 cells should be RPE-1 cells.

Reviewer #2 (Remarks to the Author):

Mooser et al provide further details for the signalling events that result in Pol I inhibition in the presence of rDNA breaks. They show that ATR has a central role in regulating Pol I transcription in response to rDNA breaks. They also show that ATR activation in the nucleoli is independent of resection- in contrast to breaks outside the nucleolus-and dependent on TOPBP1. TOPBP1 recruitment depends on Treacle and MRN complex. Deletion of TOPBP1 results in limited ATR activity and reduced transcriptional shut down. They also provide evidence that in the absence of transcriptional inhibition there is defective HDR repair at nucleolar caps. Their findings are really interesting and significantly advance both the fields of nucleolar biology and DNA damage response. There are a few additional experiments that I feel would further strengthen the evidence for their mechanism.

- 1.The authors show that ATM or ATR inhibition is sufficient to reverse the inhibition of Pol I activity after induction of rDNA breaks. Do they propose a model where the 2 kinases act epistatically?
- 2.The authors show that the DNBS1 cell line they engineered still expresses a truncated C-terminal NBS1 protein. PCR experiments can further verify this finding.
- 3.Do the authors find ATR exclusively at the periphery? Do they find ATR localising inside the nucleoli in early time points after I-Ppo1 induction? Does Treacle-TOPBP1 deletion interfere with ATR recruitment?
- 4.The authors show that resection depends on ATM and ATR activity. Do they see the same result in response to Treacle/TOPBP1 deletion? Does resection depend on Pol I inhibition?
- 5.Do the authors find pRPAS4/8-marker of extensive resection- at nucleolar periphery at late time points? What about in the DNBS1 cells?
- 6.The authors report limited nucleolar segregation in the absence of Treacle/TOPBP1. Do under these conditions identify DSBs (eg. gH2Ax staining) at the nucleolar caps? Is the lack of HDR machinery at the nucleolar caps a result of limited movement of the rDNA breaks to the periphery?
- 7.The authors previously reported that Pol I is also inhibited when damage happens outside the nucleolus and that inhibition depends on ATM activity. Do they see the same with ATR?
8. Line 221 Fig 3b instead of 2b?

Reviewers' comments:

Reviewer #1 (Remarks to the Author):

Stucki and colleagues present new data building on their previous manuscript that Treacle is required for the relocalization of TOPBP1 into the nucleolus and the activation of ATR following the induction of DNA breaks in the rDNA loci. In addition, Stucki and colleagues present evidence to suggest that both ATR and TOPBP1 are required to shut down rDNA transcription and promote nucleolar segregation.

Overall, the data is of a high quality and mostly supports the conclusions made by the authors. The cellular studies assessing the relocalisation of NBS1, Treacle and TOPBP1 to nucleoli and the biochemical studies mapping the phospho-sites on Treacle that bind TOPBP1 have been carried out to a very high standard.

Whilst the data presented does indicate a new function of Treacle in promoting the relocalisation of TOPBP1 into the nucleolus in a phospho-dependent manner, I find the story a little disjointed. It is not clear to me how ATR is activated in response to rDNA DSBs in G1 and G2, how ATR activation relates to rDNA transcriptional inhibition, why ATM and ATR are both required for rDNA transcriptional inhibition and what the relationship is between rDNA DSB repair, rDNA transcriptional inhibition, nucleolar segregation, NBS1 vs TOPBP1 recruitment to nucleoli and G2-associated resection/HR of rDNA DSBs? I appreciate that there are a lot of unanswered questions and the authors should not necessarily be required to answer them all. However, filling some of these gaps will help with the flow of the paper rather than it being a collection of interesting observations.

We would like to thank this reviewer for his/her careful evaluation of our manuscript and for assessing our data to be of high quality and to mostly support our conclusions. We were also pleased to read that this reviewer thinks that our cellular and biochemical studies have been carried out to a very high standard. Despite this positive assessment, we also appreciated the more critical comments, especially the statement that our story felt a little disjointed. We gave our best to improve upon this and added a substantial amount of new data, including an entire new main Figure describing Treacle and TOPBP1 mediated ATR recruitment in the nucleoli in response to rDNA breaks. We hope that these additions help to improve the flow of the paper.

Specific points:

1. The authors routinely use I-Ppo1 mRNA to induce DSBs in rDNA. However, the authors really need to use a fluorescent marker in their assays to show that they have actually induced comparable levels of DSBs. In addition, since the authors are directly transfecting I-Ppo1 mRNA into cells, they need to demonstrate that in each case that the levels of I-Ppo1 protein are comparable. This is particularly important when using the ATM/ATR inhibitors, given that it is possible these inhibitors are reducing the translation of the I-Ppo1 mRNA.

We thank this reviewer for raising this important point. We spent quite a lot of time and effort to clarify this issue. The I-Ppo1 expression construct we were using in this study is N-terminally fused to a V5 tag (van Sluis, M., & McStay, B. (2015). *Genes & Development*, 29, 11151-63). We therefore performed immunofluorescence experiments with V5-specific antibodies to quantify the I-Ppo1 transfection efficiency and nuclear I-Ppo1 expression levels. In 11 independent experiments with and without treatment with ATM and ATR inhibitors, we quantified the I-Ppo1 mRNA transfection efficiency by assessing nuclear V5-I-Ppo1 protein levels by quantitative imaging. This showed that while overall V5-I-Ppo1 expression levels varied from experiment to experiment, there was no pattern with regards to the inhibitor treatment (**Supplementary Fig 3b**). We also quantified V5-I-Ppo1 staining intensity per nucleus in one experiment, in which cells were pre-treated with DMSO, ATM and ATR inhibitors. No reduction of V5-I-Ppo1 expression was detectable upon inhibitor treatment, indicating that ATM and ATR inhibitors did not reduce translation of the I-Ppo1 mRNA (**Supplementary Fig 3c**). In fact, it appears as if V5-I-Ppo1 expression levels are slightly higher in ATM and ATR inhibitor treated cells as compared to DMSO treated cells. Finally, in order to rule out that the variability in the I-Ppo1 expression between individual cells would lead to over-interpretation of the data, we quantitatively assessed nucleolar segregation and TOPBP1 recruitment against V5-I-Ppo1 expression, both in the presence and absence of ATR inhibitors (**Supplementary Fig 3d**). These data reveal that very low levels of V5-I-Ppo1 expression (even beyond the immunofluorescence detection limit) induce nucleolar segregation (**Supplementary Fig 3a**). Importantly, while the I-Ppo1 expression levels do impact on the magnitude of the response (for example, high I-Ppo1 expression levels reduce the number of Treacle foci in the nucleoli more and lead to more TOPBP1 recruitment than low expression levels, the difference between ATR inhibitor treated cells and control cells is clearly detectable over the entire range of I-Ppo1 expression (**Supplementary Fig 3d**). Together, these additional data indicate that the differences observed between untreated and inhibitor treated cells are not attributable to variability in I-Ppo1 expression.

2. The authors often refer to rDNA DSBs inducing the activation of both ATM and ATR. However, to make this claim the authors really need a fluorescent marker to show that this really is the case. To me, demonstrating a reduction in rDNA transcription when the cells are treated with either ATM or ATR inhibitors is not really good enough. The authors could try using the phospho-ATM and ATR antibodies or ChIPing ATM/ATR to rDNA DSBs.

It is true that in our original manuscript, we did not present direct evidence that ATM and ATR are activated by DSB induction in the rDNA repeats, even though the effects of inhibitor treatment strongly suggested this. Moreover, we only used one selective ATR inhibitor (VE-821). We therefore repeated the nucleolar segregation assay in the presence of another ATR inhibitor (HY-19323; **Supplementary Fig 2b**) and assessed transcriptional repression and nucleolar segregation in response to I-Ppo1 expression in cells in which the ATR kinase was downregulated by siRNA (**Fig 1g** and **Supplementary Fig 2b**). ATR downregulation led to defective transcriptional repression in response to I-Ppo1 expression, and both ATR downregulation and HY-19323 treatment resulted in defective nucleolar segregation, to an extent that is comparable to VE-821 treatment. This strongly suggests that it is indeed ATR activity that is required to induce transcriptional repression and nucleolar segregation in

response to rDNA breaks. In order to directly show that ATM and ATR are activated by I-Ppo1 expression, we assessed phosphorylation of the transducer kinases CHK1 and CHK2 after I-Ppo1 transfection by Western blotting (**Fig 1e**). In response to DNA damage, CHK1 is specifically targeted by ATR on S317 and S345, and CHK2 is phosphorylated by ATM on T68. I-Ppo1 transfection led to solid phosphorylation of both CHK1 and CHK2. Inhibition of ATM abrogated CHK2 T68 phosphorylation but did not affect CHK1 phosphorylation. Inhibition of ATR significantly reduced CHK1 S317 and S345 phosphorylation but had no effect on CHK2 T68 phosphorylation. Significantly, inhibition of CHK1 signaling by two small molecule inhibitors abrogated transcriptional repression response to I-Ppo1 transfection (**Fig 1h**). Together, these new data strongly indicate that rDNA break induction by I-Ppo1 expression leads to ATM and ATR activation and that downstream signaling by CHK1 and CHK2 also significantly contributes to transcriptional suppression and nucleolar segregation.

3. I am a little confused as to why the NBS1 Δ N cells do not exhibit an ATM activation defect, when cells containing the commonly found 657Del5 NBS1 mutation does give rise to ATM activation defect.

Literature with regards to ATM activation by NBS1 hypomorphic alleles is controversial. While some studies indeed suggest that the common 657Del5 mutation does give rise to an ATM activation defect, other studies did not confirm this. Interestingly, the Nussenzweig lab showed that in a genetically defined humanized mouse model, NBS1 657Del5 did not lead to an ATM activation defect at higher doses of irradiation (8 Gy), but only at low doses of IR (2 Gy), thus suggesting a dose dependency of the response (Difilippantonio, S., et al. (2007). *The Journal of Experimental Medicine*, 204(5), 1003–1011). We performed the phosphorylation analysis at 10 Gy (**Supplementary Fig 5f**), which may explain the lack of an apparent ATM S1981 autophosphorylation defect in the NBS1 Δ N cell line. However, since we did detect a G2/M checkpoint defect in NBS1 Δ N cells (especially at lower doses of IR), we believe that ATM activation may also be compromised in the NBS1 Δ N cell line specifically at lower doses of IR. We therefore toned down the interpretation of these data in the manuscript and specifically indicated the dose of IR used.

4. The authors use RPA foci as a marker of rDNA resection. Although this indicates that there is ssDNA, it is not concrete marker of resection. The authors should use the phospho-RPA2 S4/8 antibody and native BrdU foci to provide more solid evidence that these DSBs are being resected. Is the rDNA DSB resection dependent on the nuclease activity of Mre11?

As suggested, we tested the enrichment of RPA2 pS4/8 in nucleolar caps to support our conclusion that resection is taking place in the nucleolar periphery. We also show now that RPA2 pS4/8 foci are dependent on NBS1 and MRE11 nuclease activity (**Supplementary Fig 6a**). Unfortunately, we were unable to assess resection by the native BrdU foci formation assay even though we successfully used this assay in previous studies (Anand, R et al. (2019). *The EMBO Journal*, e101005–16). The problem is that we run out of the BrdU antibody (mouse anti-BrdU from Amersham/GE Healthcare, BU-1, RPN202) used in these previous studies, and the company could not deliver a new aliquot within the revision time of this manuscript

due to shortage of stocks. Other BrdU antibodies that we tested did unfortunately not work in this assay.

5. Supplementary table 1 should contain all the specific TOPBP1 binding proteins identified by mass spec.

We have now highlighted all the known TOPBP1-interacting proteins in Supplementary table 1.

6. The authors make claims that the C-terminal phospho-domain of Treacle, which is necessary for TOPBP1 relocalisation to the nucleoli, is required for nucleolar segregation in response to I-Ppo1 expression. This is not the case. No analysis of nucleolar segregation with the Treacle S>A mutant has been shown.

This is true and we apologize for this mistake. We have now changed the text accordingly. In addition, we measured transcriptional repression in response to I-Ppo1 transfection in our GFP-Treacle expressing cell lines (WT and C-terminal phospho-mutants) that were previously treated with either control siRNA or Treacle siRNA. These data show that depletion of endogenous Treacle in GFP-Treacle WT expressing cells does not affect transcriptional repression. However, depletion of endogenous Treacle in the C-terminal phospho-domain mutants led to a significant increase in nucleolar EU incorporation, indicating that Treacle-mediated TOPBP1 relocalization in the nucleoli is required for efficient transcriptional repression in response to I-PPo1 transfection (**Fig 6d**).

7. Whilst I appreciate that it is difficult to demonstrate what the outcome of defective rDNA repair is on cell survival, the graphs depicting survival of cells depleted of Treacle, TOPBP1 or NBS1 siRNA following I-Ppo1 mRNA transfection cannot be attributed solely to defective rDNA repair, since the I-Ppo1 also induces DSBs at nuclear loci. A better approach would be to look at chromosome breaks in rDNA using metaphase-FISH, although this will not tell you what outcome a failure to repair rDNA DSBs has on cell survival.

It is true that there are about 20 I-Ppo1 recognition sites in the human genome that are not located in the rDNA repeats. However, even in the unlikely scenario that in all of these non-rDNA sites DSBs are efficiently introduced by I-Ppo1, that would still only generate a maximum of 20 DSBs per cell. Ionizing radiation generates around 40-50 DSBs per Gy, yet 1 Gy of ionizing radiation only reduces cell survival of NBS1 Δ N cells down to about 80% (**Supplementary Fig 4a**) as compared to 20% in response to I-Ppo1 transfection (**Fig 8f**). Therefore, the majority of deleterious breaks produced by I-Ppo1 are most likely originating from the nucleoli. Furthermore, Treacle is not implicated in DSB repair outside of the nucleoli, but its depletion also induces significant sensitivity to I-Ppo1 expression. We therefore believe that while I-Ppo1 sites outside of the rDNA repeats may contribute to the survival defect we observe in TOPBP1 and NBS1 deficient cells, there is enough evidence suggesting that Treacle and TOPBP1 deficient cells display an rDNA repair defect. We did try to look at the repair of chromosome breaks in rDNA using metaphase FISH as

suggested by this reviewer, but unfortunately, this assay is not straight-forward, and we failed to get robust data for it in the limited revision time.

8. Is the AAD of TOPBP1 required to initiate or maintain ATR activation in response to rDNA DSBs? What is ATR being activated by in the nucleoli in G1, when resection does not occur? Have the authors considered looking at R-loops and seeing whether this plays a role in activating ATR?

These are very intriguing questions and we tried to answer at least some of them. In order to test if the AAD of TOPBP1 is required to initiate or maintain ATR activation in response to rDNA DSBs, we would have to express a TOPBP1 AAD mutant in cells that do not express endogenous TOPBP1 and test how these cells respond to I-Ppo1 transfection. What complicates this approach is the fact that increasing the levels of TOPBP1 expression triggers transcriptional repression and nucleolar segregation even in the absence of rDNA damage (Sokka, M. et al. (2015). *Nucleic Acids Research*, 43(10), 4975–4989). The reason for this is not yet known, but interestingly, this TOPBP1 induced nucleolar response is also dependent on ATR activity. We therefore decided to use the observations of Sokka et al. to our advantage. We overexpressed GFP-tagged TOPBP1 wild type and the AAD point mutant W1145R in cells. Consistent with Sokka et al., we also observed that overexpression of wild type TOPBP1 spontaneously triggered nucleolar segregation (**Fig 7d**). Expression of the W1145R AAD mutant however failed to induce any nucleolar response. Interestingly, we also found that overexpression of TOPBP1 is sufficient to robustly accumulate ATR in nucleolar caps, even in the absence of exogenous rDNA damage (**Fig 7d**). In Treacle depleted cells, TOPBP1 overexpression is no longer triggering nucleolar segregation, indicating that Treacle-TOPBP1 complex formation in the nucleoli is the underlying mechanism of TOPBP1 overexpression-induced Pol I inhibition and nucleolar segregation. Consistent with this interpretation, it was found that TOPBP1 overexpression-induced nucleolar responses are dependent on BRCT domains 0-2 and 4-5 of TOPBP1 (Sokka et al.): these are the BRCT domains that are shown in our study to mediate interaction with Treacle (**Fig 5d**).

The question remains if Treacle-mediated TOPBP1 recruitment in the nucleoli is also essential for ATR activation under physiological conditions, i.e. in the presence of normal endogenous levels of TOPBP1. Even though we currently do not provide an answer to this question, our newly added observations that induction of rDNA breaks by I-Ppo1 leads to robust Treacle- and TOPBP1-dependent ATR accumulation in the nucleoli suggests that this may indeed be the mechanism by which ATR is being activated in the nucleoli throughout the cell cycle. Of course, further work is required to proof this concept, but we feel that this would be beyond the scope of this manuscript.

Minor point:

1. On page 21, line 479, PRE-1 cells should be RPE-1 cells.

We have corrected this mistake.

Reviewer #2 (Remarks to the Author):

Mooser et al provide further details for the signalling events that result in Pol I inhibition in the presence of rDNA breaks. They show that ATR has a central role in regulating Pol I transcription in response to rDNA breaks. They also show that ATR activation in the nucleoli is independent of resection- in contrast to breaks outside the nucleolus-and dependent on TOPBP1. TOPBP1 recruitment depends on Treacle and MRN complex. Deletion of TOPBP1 results in limited ATR activity and reduced transcriptional shut down. They also provide evidence that in the absence of transcriptional inhibition there is defective HDR repair at nucleolar caps. Their findings are really interesting and significantly advance both the fields of nucleolar biology and DNA damage response. There are a few additional experiments that I feel would further strengthen the evidence for their mechanism.

We would like to thank this reviewer for his/her evaluation of our manuscript and for considering our findings to be really interesting and a significant advance both in the field of nucleolar biology and DNA damage responses. As suggested by this reviewer, we added additional experiments to further strengthen the evidence of the mechanism we propose.

1.The authors show that ATM or ATR inhibition is sufficient to reverse the inhibition of Pol I activity after induction of rDNA breaks. Do they propose a model where the 2 kinases act epistatically?

We think the situation is similar to G2/M checkpoint activation in response to DSBs where ATM and ATR indeed act epistatically. We now show that CHK1 and CHK2 are activated upon I-Ppo1 transfection (**Fig 1e**) and that CHK1 activity is required for Pol I transcriptional inhibition in response to rDNA breaks (**Fig 1h**). Thus, we reveal that the upstream signaling events required to inhibit transcription in the nucleoli in response to rDNA breaks are remarkably similar to the upstream signaling events that arrests the cell cycle at the G2/M border.

2.The authors show that the DNBS1 cell line they engineered still expresses a truncated C-terminal NBS1 protein. PCR experiments can further verify this finding.

We used PCR to amplify the region within the NBS1 gene that was targeted by CRISPR/Cas9, cloned this region and sequenced 10 clones. This revealed the presence of three distinct indel mutations, indicating that U2OS cells contain three NBS1 alleles that were all destroyed by frameshift-inducing indel mutations (**Supplementary Fig 3b**). The truncated C-terminal NBS1 protein must therefore be expressed via an alternative translation initiation downstream of the indel mutation. We could not verify this possibility further because it would require isolation of the truncated NBS1 protein and protein sequencing by mass spectrometry. We could not do this in the limited revision time.

3.Do the authors find ATR exclusively at the periphery? Do they find ATR localising inside the nucleoli in early time points after I-Ppo1 induction? Does Treacle-TOPBP1 deletion interfere with ATR recruitment?

We now provide answers to all of these questions in the new **Fig 7**. In brief, we do find ATR also inside of the nucleoli at early time points and we show that Treacle-TOPBP1 depletion severely interferes with ATR recruitment.

4.The authors show that resection depends on ATM and ATR activity. Do they see the same result in response to Treacle/TOPBP1 deletion? Does resection depend on Pol I inhibition?

We indeed see the same result in response to Treacle and TOPBP1 depletion. These data were provided in former Fig 7a-d (now **Fig 8a-b**). We were unable to assess if resection depends on Pol I inhibition. Every treatment used (ATM/ATR inhibition, NBS1 deletion, Treacle and TOPBP1 depletion) abrogated Pol I inhibition and led to resection defects. Therefore, we did not yet identify a 'separation of function' mutant/treatment, i.e. a factor or treatment that affects only resection without having an impact on Pol I inhibition.

5.Do the authors find pRPAS4/8-marker of extensive resection- at nucleolar periphery at late time points? What about in the DNBS1 cells?

As described in the answer to Reviewer #1' s question 4, these data are now provided in **Supplementary Fig 6a**.

6.The authors report limited nucleolar segregation in the absence of Treacle/TOPBP1. Do under these conditions identify DSBs (eg. gH2Ax staining) at the nucleolar caps? Is the lack of HDR machinery at the nucleolar caps a result of limited movement of the rDNA breaks to the periphery?

We now provide answers to some of these questions in the new Fig 8: In the absence of Treacle/TOPBP1 we identified only very few DSBs by yH2AX staining in the nucleolar caps. In addition, other DSB markers such as 53BP1 were also severely reduced in Treacle/TOPBP1 depleted cells. We have not yet been able to assess whether the lack of H2AX phosphorylation and recruitment of DDR proteins in nucleolar caps is the result of limited movement of the rDNA breaks to the periphery as this would require an immuno-FISH protocol with rDNA probes. We could unfortunately not robustly establish this assay in the limited revision time.

7.The authors previously reported that Pol I is also inhibited when damage happens outside the nucleolus and that inhibition depends on ATM activity. Do they see the same with ATR?

Yes, we do indeed see the dependency also on ATR. We now provide these results in **Supplementary Fig 1d**.

8. Line 221 Fig 3b instead of 2b?

We have corrected this mistake.

Reviewers' comments:

Reviewer #1 (Remarks to the Author):

Whilst the authors have answered the majority of my comments and have significantly improved the manuscript, I would say that they have not properly addressed a couple of my issues.

1. Demonstrating that ATM and ATR are activated by I-Ppo1 expression by Western blotting does not specifically address whether these kinases are being activated by nucleolar DSBs, since I-Ppo1 induces both nucleolar and nuclear DSBs. This can only be addressed by either ChIPing ATM/ATR to rDNA DSBs or by using IF to assess foci in the nucleoli.

2. I disagree with the authors response to my concerns regarding the outcome on cell survival of not repairing nucleolar DSBs. Despite the percentage of DSBs that I-Ppo1 induces in the nucleolus relative to the nucleus, one can NOT ascribe a reduction in cell survival following I-Ppo1 expression specifically to an inability to repair nucleolar DSBs. I appreciate that metaphase-FISH is difficult and could not be carried out in time but the authors can not use colony survival assays as a biological readout of an inability to repair nucleolar DSBs when using I-Ppo1 to induce DSBs. I would suggest that the authors alter the text to acknowledge this caveat.

Reviewer #2 (Remarks to the Author):

The authors have addressed all my concerns.

Reviewers' comments:

Reviewer #1 (Remarks to the Author):

Whilst the authors have answered the majority of my comments and have significantly improved the manuscript, I would say that they have not properly addressed a couple of my issues.

We would like to thank this reviewer for his/her careful evaluation of our revised manuscript. We were pleased to read that in this reviewer's opinion our manuscript has significantly improved and that we have been able to address the majority of his/her previous concerns. In order to clarify the remaining issues, we have performed additional experiments and also slightly altered some text passages. In this rebuttal letter we answer to the two points that he/she has raised on our revised manuscript. We hope that these additions help to clarify the two remaining issues.

1. Demonstrating that ATM and ATR are activated by I-Ppo1 expression by Western blotting does not specifically address whether these kinases are being activated by nucleolar DSBs, since I-Ppo1 induces both nucleolar and nuclear DSBs. This can only be addressed by either ChIPing ATM/ATR to rDNA DSBs or by using IF to assess foci in the nucleoli.

We agree that the Western blot analysis of ATM/ATR downstream signaling events in response to I-Ppo1 expression did not yet provide enough evidence to conclude that ATM and ATR are activated by nucleolar DSBs induced by I-Ppo1 because I-Ppo1 has also a few recognition sites outside of the nucleoli. As suggested by this reviewer, we therefore carried out IF experiments with commercial ATM and ATR antibodies. The results clearly show that a fraction of endogenous ATM and ATR protein is recruited in the nucleoli at early time points after I-Ppo1 transfection (30-60 min), where it accumulates in foci that are located *inside* of the nucleoli. At later time points after I-Ppo1 transfection (120 min), these two kinases locate with the other DDR proteins in nucleolar caps (**see new Fig 1e and Supplementary Figure 2b**). At the same time, we moved the Western blot analysis (former Fig 1e) in the Supplementary Figures (**Supplementary Figure 1c**) to make space for the new IF analysis in the main Figure and to take into consideration that this analysis does not add significant evidence that ATM and ATR are activated at sites of rDNA breaks. In addition, we also removed former Fig 7a because it was redundant with new Fig 1e. We hope that these new data clarify this issue and add enough evidence to conclude that both ATM and ATR may be activated in the nucleoli.

2. I disagree with the authors response to my concerns regarding the outcome on cell survival of not repairing nucleolar DSBs. Despite the percentage of DSBs that I-Ppo1 induces in the nucleolus relative to the nucleus, one can NOT ascribe a reduction in cell survival following I-Ppo1 expression specifically to an inability to repair nucleolar DSBs. I appreciate that metaphase-FISH is difficult and could not be carried out in time but the authors can not use colony survival assays as a biological readout of an inability to repair nucleolar DSBs when using I-Ppo1 to induce DSBs. I would suggest that the authors alter the text to acknowledge this caveat.

We have rephrased the text in the relevant section to clearly state that the decreased

survival observed upon I-Ppo1 transfection in Treacle and TOPBP1 depleted cells as well as in NBS1 Δ N cells may at least partially result from defective repair at sites of I-Ppo1 recognition sites that are located outside of the nucleoli (text changes are marked in blue). We hope that this appropriately acknowledges this caveat.

Reviewer #2 (Remarks to the Author):

The authors have addressed all my concerns.

We would like to once more thank this reviewer for his/her helpful and constructive comments.

REVIEWERS' COMMENTS:

Reviewer #1 (Remarks to the Author):

The authors had adequately addressed my concerns.